# Enhancing IDS for the IoMT based on advanced features selection and deep learning methods to increase the model trustworthiness

**Ahmed Muqdad Alnasrallah**[1]*, **Maheyzah Md Siraj**[2]*, **Hanan Ali Alrikabi**[3]

**1** Faculty of Education for Pure Sciences, University of Thi-Qar, Nasiriyah, Iraq, **2** Faculty of Computing, Universiti Teknologi Malaysia, Johor, Malaysia, **3** Marshlands Research Center, University of Thi-Qar, Nasiriyah, Iraq

* ahmed@utq.edu.iq (AMA); maheyzah@utm.my (MMS)

**Editor:** mamoona humayun, University of Roehampton, UNITED KINGDOM OF GREAT BRITAIN AND NORTHERN IRELAND

## Abstract

Information technology has significantly impacted society. IoT and its specialized variant, IoMT, enable remote patient monitoring and improve healthcare. While it contributes to improving healthcare services, it may pose significant security challenges, especially due to the growing interconnectivity of IoMT devices. Hence, a robust IDS is required to handle these issues and prevent future intrusions in a appropriate time. This study proposes an IDS model for the IoMT that integrates advanced feature selection techniques and deep learning to enhance detection performance. The proposed model employs Information Gain (IG) and Recursive Feature Elimination (RFE) in parallel to select the top 50% of features, from which intersection and union subsets are created, followed by a deep autoencoder (DAE) to reduce dimensionality without losing important data. Finally, a deep neural network (DNN) classifies traffic as normal or anomalous. The Experimental results demonstrate superior performance in terms of accuracy, precision, recall, and F1 score. It achieves an accuracy of 99.93% on the WUSTL-EHMS-2020 dataset while reducing training time and attains 99.61% accuracy on the CICIDS2017 dataset. The model performance was validated with an average accuracy of 99.82% ± 0.16% and a statistically significant p-value of 0.0001 on the WUSTL-EHMS-2020 dataset, which refers to stable statistical improvement. This study indicates that the proposed strategy decreases computational complexity and enhances IDS efficiency in resource-constrained IoMT environments.
The following Table 1 shows the abbreviations used in this paper.

## Introduction

The rapid improvements have greatly improved the quality of life in information technology, especially the Internet of Things (IoT), which has integrated into a wide range

**Data availability statement:** The study utilizes publicly available datasets: CICIDS2017: https://www.unb.ca/cic/datasets/ids-2017.html. WUSTL-EHMS-2020 dataset: https://www.cse.wustl.edu/~jain/ehms/index.html. The code is accessible at: https://github.com/Ahmed85iq/IoMT-IDS-DL.

**Funding:** The author(s) received no specific funding for this work.

**Competing interests:** The authors have declared that no competing interests exist.

of areas, from basic applications to intricate systems like smart cities and hospitals. The IoT is a network of linked systems, sensors, and gadgets that provide users with a wide range of technological services [1]. The IoMT is a crucial and extensively investigated application area within the Internet of Things. By automating and managing healthcare services, IoMT transforms the medical field and improves the precision and efficiency of medical equipment and services [2]. Wearable IoMT smart devices, permit remote, enabling real-time patient monitoring, fast diagnosis, and healthcare delivery [3].

Due to IoMT directly affecting people's lives, security must be guaranteed. IoMT network breaches can cause serious health hazards and even put lives in danger by interfering with medical examinations and diagnoses, preventing patients and healthcare providers from communicating or resulting in the loss or theft of medical data. The healthcare industry has been the subject of an increase in cyberattacks in recent years, according to statistics [4]. While the IoMT enhances medical services for patients, it also presents a substantial threat to healthcare providers due to the persistent possibility of cyberattacks, which can result in extensive harm. A Distributed Denial of Service (DDoS) attack poses a significant risk to the healthcare system by causing disruption to linked devices and rendering medical services inaccessible to authorized customers [5]. Man-in-the-middle is a form of cyberattack when an

**Table 1. List of Abbreviations and Symbols.**

| Abbreviation | Definition | Abbreviation | Definition |
|---|---|---|---|
| IoT | Internet of Things | IoMT | Internet of Medical Things |
| IDS | Intrusion Detection System | IG | Information Gain |
| RFE | Recursive Feature Elimination | DAE | Deep Autoencoder |
| DNN | Deep Neural Network | WUSTL-EHMS-2020 | Washington University in St. Louis Electronic Health Monitoring System 2020 Dataset |
| ML | Machine Learning | CICIDS2017 | Canadian Institute for Cybersecurity Intrusion Detection System 2017 Dataset |
| DL | Deep Learning | ACC | Accuracy |
| PRE | Precision | DR | Detection Rate |
| FNR | False Negative Rate | FPR/FAR | False Positive Rate/ False Alarm Rate |
| ICI | Incorrect Classification Instances | ER | Error Rate |
| MCC | Matthew's Correlation Coefficient | AUC | Area Under the Curve |
| TP | True Positive | TN | True Negative |
| FP | False Positive | FN | False Negative |
| ReLU | Rectified Linear Unit (activation function) | TFSIG | Top Fifty Percent-Feature Subset based on Information Gain |
| TFS-U | Union of TFSIG and TFSRFE | TFSRFE | Top Fifty Percent-Feature Subset based on Recursive Feature Elimination |
| TFS-I | Intersection of TFSIG and TFSRFE | HIPAA | Health Insurance Portability and Accountability Act (US data privacy regulation) |
| DoS | Denial of Service | GDPR | General Data Protection Regulation (EU data privacy regulation) |
| DDoS | Distributed Denial of Service | MitM | Man-in-the-Middle Attack |
| PSO | Particle Swarm Optimization | | |

adversary intercepts communication between IoMT devices to manipulate and steal sensitive data. This attack greatly affects the trustworthiness and secrecy of patient data. Ransomware is a different kind of hack that poses a threat to the healthcare system. In this scenario, a malicious entity encrypts IoMT systems, thereby impeding healthcare professionals and stakeholders from accessing medical records and delivering healthcare services to patients [3,6].

To overcome these risks and ensure the security and privacy-preserving of information exchange in the IoMT network, innovative and efficient feature selection techniques are urgently required for IoMT data processing. These techniques are essential for supporting (IDS) that are dependable to detect Anomalies in traffic. IoMT's high-dimensional, feature-rich network traffic needs in-depth analysis via deep learning and machine learning models. This can be laborious and complicated, which lowers the effectiveness of IDS. Consequently, to improve the speed and effectiveness of IDS against Cybersecurity attacks, the feature set must be shrunk to concentrate on the most important aspects. Furthermore, the process of decreasing dimensions and removing superfluous characteristics results in the creation of a lightweight IDS, this is crucial in effectively and efficiently addressing all forms of threats in IoMT networks.

The proposed approach reduces IDS complexity while ensuring stability and accuracy. This is achieved by combining feature selection and dimension reduction with a proper classifier. This model is designed to efficiently detect both common and rare threats. The contributions of the proposed innovative approach in this paper can be summarized as follows:

1. Focus on combining IG and RFE to select the top 50% features to enhance performance and reduce complexity.

2. Dimensionality reduction of features using a DAE to enhance accuracy and maintain data privacy.

3. In-depth evaluation by critical datasets; the WUSTL-EHMS-2020 dataset is relevant in the IoMT area and the CICIDS-2017 is a comprehensive dataset designed for evaluating IDS.

This paper presented an approach that outperforms earlier approaches in key metrics: accuracy, specificity, and sensitivity. It improved training efficiency and reduced prediction time, making it possible for real-time computing environments in IoMT networks. The remainder of this paper is structured as follows; the Related Work section presents a comprehensive review of relevant literature. The Proposed Model section describes the IDS framework, including data preprocessing, feature selection, dimensionality reduction, and classification. The Experimental Results and Analysis section outlines the datasets and configurations used. The Results and Discussion section reports and analyzes the evaluation results. The Comparison section contrasts the proposed model with state-of-the-art approaches. The Conclusion summarizes the findings, and the Future Work section outlines directions for further research.

## Related work

The IoMT is the confluence of medical devices with IoT technology that improves patient monitoring, streamlines healthcare delivery, and broadens data collection [7]. However, increased interconnectivity and information exchange raise the threat index of IoMT systems about cybersecurity to extremely high. Intrusion detection takes great importance in an IoMT environment for its attack detection and mitigation capabilities. Efficient IDS creation is of paramount importance for ensuring the security and confidentiality of sensitive medical information. Numerous studies have concentrated on developing IDS for conventional networks, although there is a dearth of research on IDS specifically designed for the IoMT [8]. This literature review stands for current trends in IDS for the IoMT, outlining the machine learning and deep learning methods to enhance in accuracy and efficiency of detection.

IoMT consists of interconnected medical equipment, sensors, and applications. This network generates large volumes of data and meets distinct cybersecurity obstacles at the perception, network, and application layers. The perception layer provides secure gathering and transmission of data, the network layer protects the integrity and confidentiality of data, and the application layer ensures strong access control and protection of privacy. Conventional security measures based on rules are not sufficient for the constantly changing and diverse nature of the IoMT, which requires more flexible

and intelligent IDS solutions [6]. In the context of the IoMT IDS are utilized [9]. Several feature selection techniques have been employed in previous studies to decrease the dimensionality of input characteristics while maintaining their pertinent information [10]. Filter-based techniques, such as chi-square and information gain (IG), assess the impact of each feature on the target variable individually, wrapper-based approaches, such as recursive feature elimination (RFE), employ machine learning and deep learning algorithms to systematically choose and eliminate features based on their influence on the model's performance [8,11,12]. This approach quantifies the individual impact of each attribute on the model's prediction. Hence, the integration of wrapper feature selection techniques with other machine learning or deep learning algorithms is employed to develop a highly efficient IDS for the IoMT. Random Forest, like other embedded approaches, incorporates feature selection into the training process [13]. Feature-based information Selection is a filter-based technique that employs information theory principles, such as entropy and mutual information, to determine the significance of features. This methodology assesses the correlation between the characteristics and the target variable and exclusively chooses the most informative characteristics [14,15]. Applying feature selection algorithms based on maximum relevance and minimum redundancy to select the significant features, this is used in many fields, including IDS in the Internet of Things [16]. Feature selection techniques based on information theory are very suitable and useful for enhancing IDS in the IoMT. These techniques are capable of efficiently managing the high-dimensional and intricate data that is frequently encountered in this field [16]. Based on Mutual Information The process of Feature Selection encounters difficulties due to a lack of sufficient data, resulting in inaccurate feature selection. Methods such as data augmentation and transfer learning can be employed to address the issue, while they may potentially create biases or incur additional computing expenses. Another strategy aims to achieve a balance between redundancy and relevancy. therefore, the DAE can be utilized to decrease the dimension of the selected features [17]. This ultimately enhances the effectiveness of the proposed approach in extracting the most valuable information from the selected features. This paper [18] presents a hybrid Fast (APSO-WOA-CNN) method to improve IDS in IoT environments to improve IDS accuracy and reduce false positives. The new model outperformed previous models, making it a potent IDS improvement tool. The authors of this study [19] show a new algorithm that is a mix of Fast R-CNN and Gradient Descent Boosting (GBR). This algorithm is meant to make IDS more accurate in IoT settings and cut down on false positives when looking for DoS and DDoS attacks. The results reflect that the proposed model outperforms traditional models and hence can be a strong tool for the improvement of IDS.

Recent literature, which focuses on innovative approaches to enhanced accuracy and operational efficiency, provides a deeper understanding of IDS in IoMT environments. For example, a study [20] presented an IDS model based on Vision Transformers, demonstrating how modern deep learning techniques can be used to efficiently process complex data. The research [21] focused on developing models capable of detecting zero-day attacks using machine learning techniques, reflecting the importance of generalization when dealing with unknown or unexpected data. Furthermore, the study [22] conducted a comprehensive review and detailed analysis of the challenges associated with using artificial intelligence to detect zero-day attacks, recommending the use of hybrid models and dimensionality reduction techniques to improve performance.

Finally, the study [23] demonstrated the potential for designing networks with attention-based architectures to achieve better integration of time-sensitive features, providing valuable insights applicable to highly sensitive IoMT environments.

These studies provide a sufficient number of insights and ideas that bring forth the diversity and effectiveness of various algorithms and models against the rising threat of cyber-attacks. The proposed approach in this research assesses the efficacy of the IDS within the IoMT framework by incorporating assessment criteria that extend beyond conventional accuracy measures. The measures encompass detection rate (DR), false negative rate (FNR), false positive rate (FPR), incorrect classification instances (ICI), error rate (ER), as well as accuracy (ACC), precision (PRE), and Matthew's correlation coefficient (MCC). This comprehensive assessment provides a more thorough understanding of the system's performance. The state-of-the-art ML and DL-based methods for IoMT attack detection are compared in Table 2.

**Table 2. Recent studies of attack detection using ML and DL in IoMT.**

| Article | Dataset | Technique | Limitation |
|---|---|---|---|
| [24] | CICIDS2017 | RF | Only network traffic features, not focused on IoMT network. |
| [25] | ToN-IoT | CNN-LSTM | Dataset not related with IoMT, as well as only focus on the malware attack detection |
| [26] | ToN-IoT | Swarm NN | Only network traffic features, Dataset not related with IoMT. |
| [17] | WUSTL EHMS 2020 | PSO- DNN | The classification accuracy can be improved. |
| [8] | WUSTL EHMS 2020 | LRGU-MIFS with SVM, LR, RF, DT, and LSTM | The classification accuracy can be further improved. |
| [27] | WUSTL EHMS 2020 | RFE-DT, Ridge Regression -CNN and LSTM | The Ridge-CNN model has a higher FAR, which suggests the presence of potential misclassification problems. |
| [28] | WUSTL EHMS 2020 | DNN – XMeDNN | The classification accuracy can be improved. |
| [29] | ToN-IoT | Ad-Boosting, DT, RF | Dataset not related with IoMT |

## Proposed model

The proposed IDS for IoMT consists of three main components: data pre-processing, feature selection, and classification, as shown in Fig 1. In IoMT contexts, network traffic is frequently characterized by noise, including NaN values and missing data, as well as a multitude of redundant characteristics in both textual and numerical formats. To tackle these problems, the initial stage entails data pre-processing, wherein the network traffic data is cleansed by eliminating superfluous textual information, substituting NaN, and missing values with zeros. The purpose of this stage is to streamline the dataset for future feature selection and IDS operations.

The main strategies for feature selection included Information Gain (IG) and Recursive Feature Elimination (RFE). Information gain was selected to evaluate the characteristics based on ranking their importance and their classification role, while RFE recursively removes fewer key features to keep the most influential ones. The 50% feature ratio was chosen based on experimental validation, as several different ratios were chosen and assessed, and the 50% ratio was proved effective in reducing model complexity, stability, and overfitting.

IG and RFE have been chosen due to their relative simplicity and efficiency in processing intricate datasets, such as Electronic Health Monitoring System 2020 (WUSTL-EHMS-2020) and the Canadian Institute for Cybersecurity Intrusion Detection System 2017 (CICIDS2017), hence offering an optimal mix of accuracy, complexity, and computational cost reduction. These techniques guarantee feature interpretability and provide enabling for IoMT applications. The system uses filter-based techniques like IG and RFE to determine the top 50% of ranked features from the refined dataset. The chosen characteristics are separated into two subsets: the Top Fifty Percent-Feature Subset based on IG (TFSIG) and the Top Fifty Percent-Feature Subset based on RFE (TFSRFE). By applying intersection and union procedures to these sub-sets, two novel subsets of features (NFS), TFS-I (intersection) and TFS-U (union) are generated [1]. Therefore, the DAE can be employed to reduce the dimensionality of the chosen features.

The classifier uses the new feature subsets, TFS-I with DAE and TFS-U with DAE, as their inputs. The Deep Neural Network (DNN) separates the dataset into two categories: normal (BENIGN) or abnormal (ATTACK) traffic. The system's performance is evaluated by key metrics, including ACC, PRE, DR, FNR, FPR, ICI, ER, MCC, and Memory. The purpose is to select a subset of features that reduces the total number while enhancing accuracy, precision, and detection rate relative to the original feature set.

 

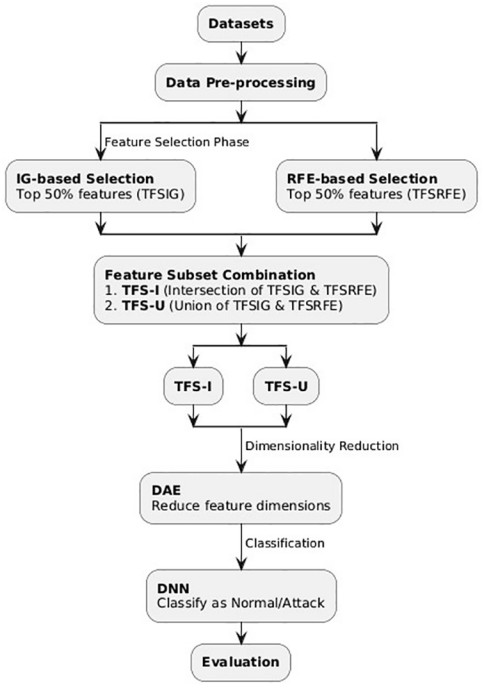

**Fig 1. Proposed model framework.**

## Features selection and reduction approach

The process of selecting features is crucial in constructing an AI model. This procedure aims to construct a subset of representative features from the original dataset, selected features ensure that are useful and significantly impact the classification process, as well as to decrease computing complexity by concentrating just on the most noteworthy features.

Recent studies show that feature selection strategies improve IDS efficiency and accuracy in IoT contexts. Effective model complexity, accuracy, and training time reduction are essential for resource-constrained IoT systems that require fast and dependable responses. Similarly, the study [30], combining RFE-RIDGE with only 13 important features, improved classification performance to 97.85% accuracy and 0.03 false alarm rate. Reduced data dimensionality enhanced computational performance and complexity. The study [31] suggested IGRF-RFE, a hybrid technique for choosing key features. Selecting 23 of 42 UNSW-NB15 characteristics boosted classification accuracy to 84.24%. These approaches have increased computing efficiency and reduced complexity while preserving IDS performance. In [8], the authors presented LRGU-MIFS feature selection. This method calculates feature recurrence, decreasing feature duplication. Only 30 dataset features were picked using this method. This method improves performance and computational complexity, achieving 95.5% classification accuracy in IoMT.

Considering these results, the innovative approach to this work, which combines IG and RFE, using intersection and union methods, and DAE for dimensionality reduction, is effective and applicable in IoMT contexts. It balances high performance and computational efficiency while lowering feature size to the smallest set without compromising model accuracy. These features, including biometric and network flow data, are critical for detecting IoMT-specific threats such as spoofing or data injection.

## Information gain (IG)

IG is a commonly employed method for selecting features in artificial intelligence that is based on the concept of information entropy. The method is especially efficient for datasets that have a large of dimensions and features that are not

evenly distributed, as these factors can have a detrimental effect on the performance of deep learning models. IG quantifies the significance of each feature by computing its information entropy and assigning a numerical value ranging from zero to one. A value of zero signifies the absence of information, while a value of one shows the highest possible amount of information as the Fig 2. According to this criterion, IG chooses features with the greatest information values to create the necessary subset [32].

### Recursive feature elimination (RFE)

RFE is a technique used to choose the most key features in a dataset by recursively eliminating futile features. RFE is an iterative method for selecting features that try to identify the most crucial characteristics in a dataset. It eliminates the least essential elements depending on their impact on the model's performance. RFE improves the effectiveness and productivity of deep learning and machine learning models by pinpointing a subset of the noteworthy features. This approach aids in reducing overfitting and has a favourable influence on the classification procedure as Fig 3. The features are prioritised according to their significance, and the process continues until the specified number of features is attained [33].

### Dimensions reduction with deep autoencoder

The Auto-Encoder is a Neural Network design that has been recently investigated for its ability to perform feature selection. A neural network generally consists of three major layers: input, hidden, and output, as shown in Fig 4. The input layer will take the features of a connection, while the output layer provides the class label of the connection. The hidden layer of the neural network will handle the process of analysing the features and finding the underlying relationship between them. AEs are neural networks that are unsupervised and attempt to find a lower-dimensional representation of the data, they do so by feeding input features into the network and then generating the same features as output. The process involves encoding and decoding input, using backpropagation to change the weights and minimize error. The hidden neuron values are used as the reduced feature space. In this work, a symmetrical autoencoder was used with an equal number of neurons for both encoder and decoder components [34,35].

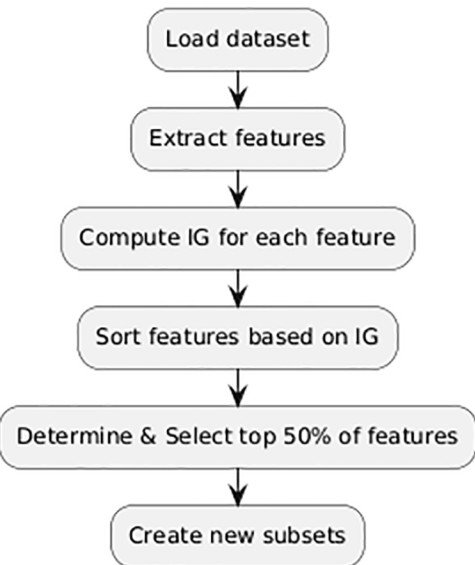

**Fig 2. IG feature selection process.**

## Feature Selection using RFE

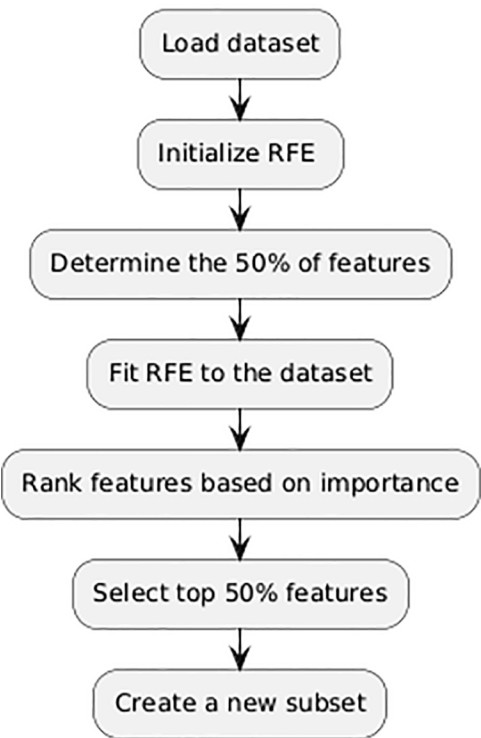

**Fig 3. RFE feature selection process.**

## Basic Autoencoder Architecture

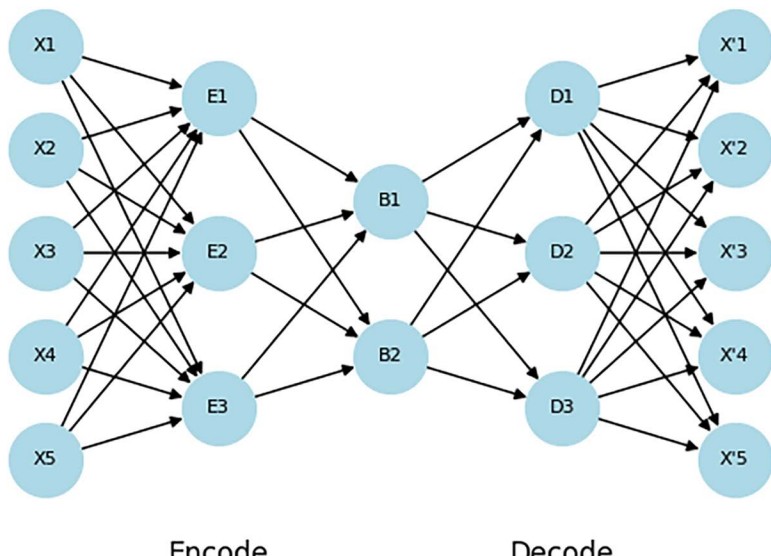

**Fig 4. Shows the basic architecture of an autoencoder.**

DAE helps reduce dimensionality without losing essential features, making the model lighter and faster to train and execute and increasing the model's focus on the patterns most relevant for classification.

## Classification process with deep neural networks

The data classification stage is crucial for assessing the efficacy of the proposed model in differentiating or forecasting malevolent data from the extensive volume of network traffic data. This paper presents the Deep Neural Network (DNN) as the classifier and evaluates their respective results.

DNN, a form of artificial neural network employed in deep learning, have three primary layers: the input layer, hidden layers (often one or more), and the output layer as the Fig 5. The layers consist of interconnected nodes referred to as neurons. DNNs are highly effective tools for identifying network breaches. They can precisely classify traffic patterns as regular or malicious, even when managing large datasets [36]. This study's DNN model is developed to meet the demands of IDS in IoMT, addressing the challenges of high dimensionality and data complexity. Three concealed layers comprising 64, 32, and 16 neurons utilize ReLU activation to process non-linear inputs. This quantity of neurons was selected to minimize computing complexity and extract essential information. The final layer comprises a single neuron utilizing the Sigmoid activation function for binary classification (BENIGN or ATTACK). The learning rate was set to 0.001 with the Adam optimizer, and batch size was 256, both carefully tuned in order to balance performance and complexity. Early Stopping and Reduce Learning Rate on Plateau enhanced model stability and reduced overlearning. This architecture provides perfect accuracy for two datasets.

The integration of DAE and DNN with selected features as a key innovation has positively impacted the design of IDS in IoMT environments, allowing the model to focus on the most significant features that showed high classification accuracy while reducing computational complexity, overfitting, scalability and protection of sensitive data.

## Performance metrics

Various essential performance measures are employed to assess the efficacy of the proposed framework [1,37,38].
**Confusion matrix:** It is a Table that summarizes the performance of a classifier, every column is a sample of a predicted class, and every row is an actual class. It contains information about true positives (TP), true negatives (TN), false positives (FP), and false negatives (FN).

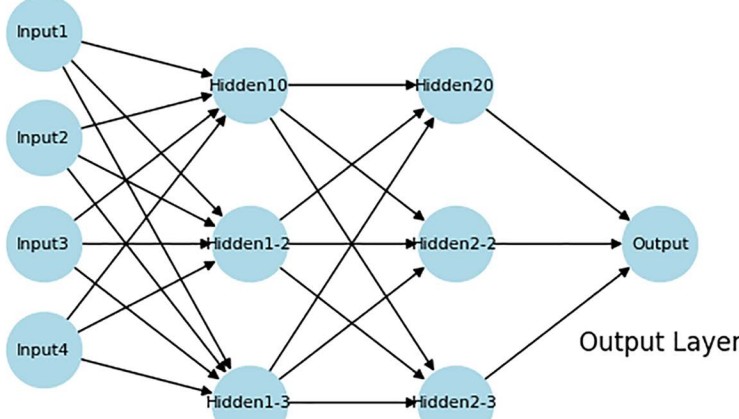

**Basic DNN Architecture for Binary Classification**

**Fig 5. Architecture of deep neural networks.**

**Accuracy (ACC):** refers to the amounts of correctly classified instances divided into the total instances. Calculated by Equation 1.

$$Acc = \frac{(TP + TN)}{(TP + TN + FP + FN)}$$

(1)

**Precision (PR):** It is known (positive predictive value) too, refers to measures of the accuracy of positive predictions. calculated by Equation 2.

$$PR = \frac{TP}{(TP + FP)}$$

(2)

**Detection Rate (DR)** It is known (True Positive Rate or Sensitivity) too, measures the amount of true positive instances that are correctly identified. Calculated by Equation 3.

$$DR = \frac{TP}{(TP + FN)}$$

(3)

**False Negative Rate (FNR):** measures the amount of true positive instances that are incorrectly classified as negative. Calculated by Equation 4.

$$FNR = \frac{FN}{(FN + TP)}$$

(4)

**False Alarm Rate (FAR):** It is known (False Positive Rate or Type I Error) too, measure the amount of true negative instances that are classified as positive incorrectly. Calculated by Equation 5.

$$FAR = \frac{FP}{(FP + TN)}$$

(5)

**Incorrect Classification Instances (ICI):** Represent the total number of misclassified instances, including false positives and false negatives. Calculated by Equation 6.

$$ICI = FP + FN$$

(6)

**Error Rate (ER):** Refers to the number of misclassified instances divided to the total instances. Calculated by Equation 7.

$$ER = \frac{(FP + FN)}{(TP + TN + FP + FN)}$$

(7)

**UC-ROC:** The AUC ROC curve shows classification model accuracy and error rates. A classifier model's performance is quantified using the Receiver Operating Characteristics Curve (ROC curve). The ROC curve shows the classifier model's ability to correctly identify positive instances by plotting the true positive rate and false positive rate. AUC measures a classifier's ability to discern classes. It often summarizes the ROC curve. Higher AUC models are better at discriminating positive and negative classes, indicating better accuracy [39].

**MCC (Matthew's correlation coefficient):** Performance metric is a quantitative measure used to assess the effectiveness of binary classification models in machine learning, as defined by Equation (8). Furthermore, the values of 1 and +1 represent a flawless misclassification and classification, respectively. Conversely, a magnitude of MCC = 0 corresponds to the anticipated outcome for a random or "coin-tossing" classifier [27].

$$MCC = \frac{(TP * TN - FP * FN)}{\sqrt{((TP + FP) * (TP + FN) * (TN + FP) * (TN + FN))}} \tag{8}$$

**Memory:** This is a measure of the model's memory consumption while it is running (MB and GB units of measurement) [27].

### Dataset overview

This study investigated models using real network traffic from WUSTL-EHMS-2020 and CICIDS2017 datasets. Using these datasets ensures the anomaly detection system's scalability and ability to manage a variety of IoMT assaults and network behaviour. This model may be deployed in different contexts and evaluated on balanced and unbalanced datasets using these two datasets. By choosing multiple datasets, one may assess how well the proposed approach generalises across network architectures.

### WUSTL-EHMS-2020 dataset

The healthcare dataset used to assess the effectiveness of the developed approaches is called WUSTL-EHMS-2020. The WUSTL-EHMS-2020 dataset includes both network flow and patient biometric data. The dataset includes many categories of cyberattacks, including MitM attacks, data injection, and spoofing. Table 3 and Fig 6, present statistical information extracted from the WUSTL-EHMS-2020 dataset. The dataset has 44 features, with 35 being network flow data, eight being biometric features derived from patients' data, and one being a label feature [40].

### CICID2017 dataset

The CIC IDS 2017 dataset is highly recommended for Anomaly detection system algorithms because to its inclusion of up-to-date benign and common attack data that closely mimic real-world situations. This dataset contains the outcomes of the CIC Flow Metre network traffic assessment, Table 4, and Fig 7, which present statistical information extracted from the CICID2017 dataset where flows are categorized based on the date, source and destination IP addresses, port, protocol, and type of attack. The collection includes of 84 features, each accompanied by its corresponding traffic status. The CICIDS2017 dataset encompasses a range of malevolent network activity and risks, including brute force attacks, heartbleed attacks, botnet attacks, denial of service attacks, and web-based attacks [4].

This study uses WUSTL-EHMS-2020, which includes biometric and network data from medical devices, for IoMT. CICIDS2017, a comprehensive dataset that encompasses attacks across various IoT scenarios, was utilized to broaden the evaluation. By integrating the two datasets, the model can generalize across contexts and test the system's ability to respond to complex and diverse challenges. Recent research like [41] and [42] used the CICIDS2017 dataset to evaluate IoMT security systems, supporting this methodology.

**Table 3. WUSTL-EHMS-2020 dataset information.**

| Measurement | Value |
|---|---|
| Dataset size | 12.96 MB |
| No. of normal samples | 14272 (87.46%) |
| No. of attack samples | 2046 (12.54%) |
| Total number of samples | 16318 |

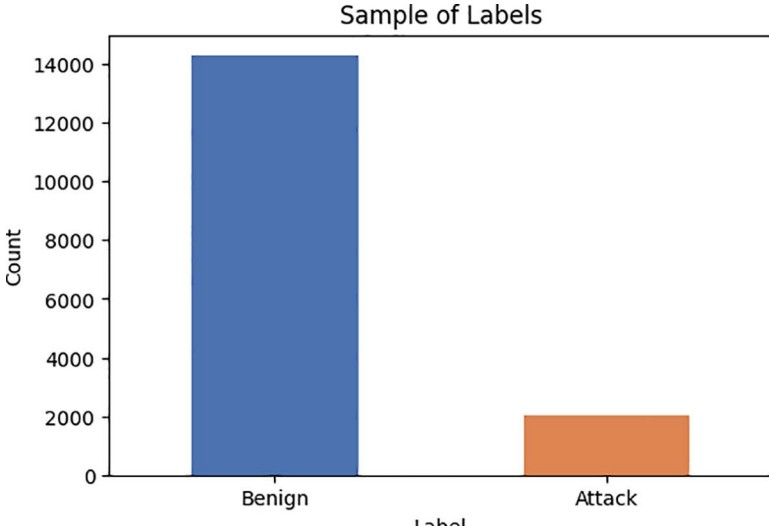

**Fig 6.** **Represent the number of attacks to the number of benign in WUSTL-EHMS-2020 dataset.**

**Table 4.** **CICID2017 dataset information.**

| Measurement | Value |
|---|---|
| Dataset size | 2.88 GB |
| No. of normal samples | 2273097 (72.87%) |
| No. of attack samples | 846248 (27.13%) |
| Total number of samples | 3119345 |

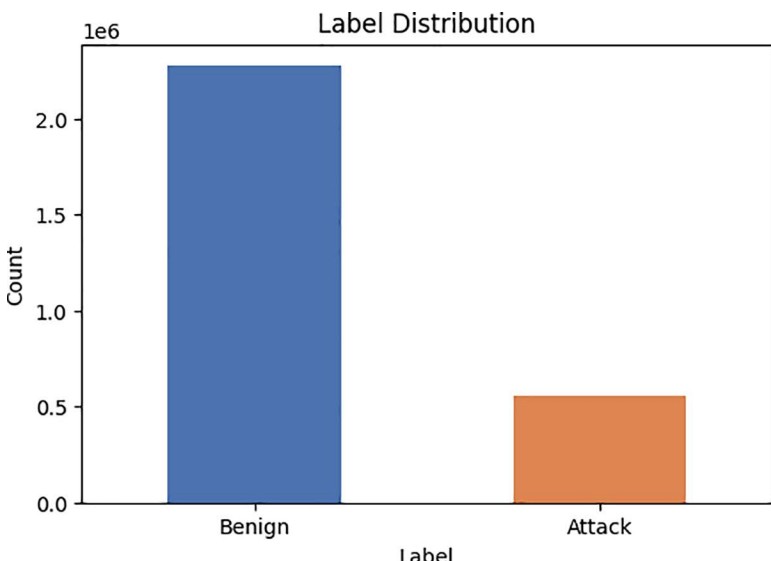

**Fig 7.** **Represent the number of attacks to the number of benign in CICID2017 dataset.**

## Experimental results and analysis

The WUSTL-EHMS-2020 and CICIDS2017 datasets were used to test the proposed model in this study, and it was compared with recent works. The datasets have 44 and 80 features, respectively, including anomaly and regular data.

The research consists of two primary stages: feature selection and classification.

During the feature selection phase, both IG and RFE were employed to individually select the top 50% of features in the dataset. This process yielded two distinct subsets of features: the Top Fifty Percent-Feature Subset based on Information Gain (TFSIG) and the Top Fifty Percent-Feature Subset based on Recursive Feature Elimination (TFSRFE), then applied the DAE to selected features to make dimension reduction of subgroups. Through the execution of intersection and union operations on the two subgroups, two further subgroups were derived: TFS-I (Intersection of TFSIG and TFS-RFE) and TFS-U (Union of TFSIG and TFSRFE). Four distinct sets of features (TFSIG, TFSRFE, TFS-I, TFS-U) were generated by this procedure, as illustrated in Tables 5 and 6.

During the data analysis and categorization stage, the DAE was applied to dimensions reduction and DNN was utilized to evaluate the four subgroups individually to identify the most optimal subset. The performance evaluation included the measurement of accuracy (ACC), precision (PRE), detection rate (DR), false negative rate (FNR), false positive rate (FPR/FAR), erroneous classification instances (ICI), and error rate (ER), as presented in Tables 7 and 8. The most optimal outcomes were obtained by utilizing DNN on the TFS-U subset with both datasets under research WUSTL-EHMS-2020 and CICIDS2017, where excellent results were obtained from each of them, for example, the accuracy was 99.9387 and 99.6121, respectively, from this TFS-U subset.

Table 5. Selected Features via a proposed approach from WUSTL-EHMS-2020.

| Methods | Selected features | No. of features |
|---|---|---|
| Original Features | 0, 1, 2, 3, 4, 5, 6, 7, 8, 9, 10, 11, 12, 13, 14, 15, 16, 17, 18, 19, 20, 21, 22, 23, 24, 25, 26, 27, 28, 29, 30, 31, 32, 33, 34, 35, 36, 37, 38, 39, 40, 41, 42, 43 | 44 |
| TFSIG | 1, 4, 7, 8, 9, 12, 13, 16, 17, 22, 25, 26, 31, 32, 34, 35, 36, 37, 40, 41, 42, 43 | 22 |
| TFSRFE | 1, 2, 3, 5, 6, 7, 8, 9, 10, 11, 14, 15, 16, 21, 25, 26, 27, 28, 30, 31, 32, 36 | 22 |
| TFS-U | 1, 2, 3, 4, 5, 6, 7, 8, 9, 10, 11, 12, 13, 14, 15, 16, 17, 21, 22, 25, 26, 27, 28, 30, 31, 32, 34, 35, 36, 37, 40, 41, 42, 43 | 34 |
| TFS-I | 32, 1, 36, 7, 8, 9, 16, 25, 26, 31 | 10 |

Table 6. Selected Features via a proposed approach from CICIDS2017.

| Methods | Selected features | No. of features |
|---|---|---|
| Original Features | 0, 1, 2, 3, 4, 5, 6, 7, 8, 9, 10, 11, 12, 13, 14, 15, 16, 17, 18, 19, 20, 21, 22, 23, 24, 25, 26, 27, 28, 29, 30, 31, 32, 33, 34, 35, 36, 37, 38, 39, 40, 41, 42, 43, 44, 45, 46, 47, 48, 49, 50, 51, 52, 53, 54, 55, 56, 57, 58, 59, 60, 61, 62, 63, 64, 65, 66, 67, 68, 69, 70, 71, 72, 73, 74, 75, 76, 77, 78, 79 | 80 |
| TFSIG | 1, 2, 3, 6, 7, 8, 10, 11, 12, 13, 14, 15, 16, 17, 18, 20, 22, 23, 24, 25, 30, 37, 38, 39, 41, 42, 43, 44, 48, 49, 54, 55, 56, 65, 67, 68, 69, 76, 78, 79 | 40 |
| TFSRFE | 1, 2, 5, 6, 7, 8, 10, 11, 12, 14, 15, 17, 18, 19, 20, 21, 22, 23, 24, 25, 26, 32, 34, 38, 41, 42, 43, 44, 46, 51, 54, 55, 56, 64, 65, 66, 67, 77, 78, 79 | 40 |
| TFS-U | 1, 2, 3, 5, 6, 7, 8, 10, 11, 12, 13, 14, 15, 16, 17, 18, 19, 20, 21, 22, 23, 24, 25, 26, 30, 32, 34, 37, 38, 39, 41, 42, 43, 44, 46, 48, 49, 51, 54, 55, 56, 64, 65, 66, 67, 68, 69, 76, 77, 78, 79 | 51 |
| TFS-I | 1, 2, 6, 7, 8, 10, 11, 12, 14, 15, 17, 18, 20, 22, 23, 24, 25, 38, 41, 42, 43, 44, 54, 55, 56, 65, 67, 78, 79 | 29 |

**Table 7. Model performance results based on the WUSTL-EHMS-2020 dataset.**

| Method | ACC % | PR% | REC% | F1% | DR % | FNR | FPR/ FAR | ICI % | Error Rate | AUC | MCC | Titmice/ sec. |
|---|---|---|---|---|---|---|---|---|---|---|---|---|
| Original Features | 88.786 | 55.364 | 62.019 | 40.716 | 0.6202 | 0.379 | 0.0730 | 29.867 | 0.1121 | 79.9857 | 52.16 | 12.84 |
| TFSIG | 90.196 | 87.500 | 26.923 | 41.176 | 26.92 | 0.730 | 0.0056 | 3.3866 | 0.0980 | 69.4658 | 64.08 | 12.52 |
| TFSRFE | 92.861 | 94.202 | 46.875 | 62.600 | 46.88 | 0.531 | 0.0042 | 2.7315 | 0.0714 | 99.3874 | 89.66 | 7.52 |
| TFS-I | 98.805 | 97.243 | 93.269 | 95.214 | 93.27 | 0.067 | 0.0039 | 2.5819 | 0.0119 | 99.8831 | 96.25 | 4.43 |
| **TFS-U** | **99.938** | **99.521** | **100** | **99.760** | **99.76** | **0** | **0.0007** | **0.4788** | **0.0006** | **100** | **99.59** | **3.2** |

**Table 8. Model performance results based on the CICIDS2017 dataset.**

| Method | ACC % | PR% | REC% | F1% | DR % | FNR | FPR/ FAR | ICI % | Error Rate | AUC |
|---|---|---|---|---|---|---|---|---|---|---|
| Original Features | 97.3703 | 93.2020 | 93.4900 | 93.3457 | 98.79 | 0.0651 | 0.0168 | 6.4645 | 0.0263 | 99.6667 |
| TFSIG | 98.4758 | 93.0785 | 99.6874 | 96.2697 | 99.69 | 0.0031 | 0.0182 | 7.0279 | 0.0152 | 99.9167 |
| TFSRFE | 99.0505 | 96.2309 | 99.0676 | 97.6286 | 99.07 | 0.0093 | 0.0095 | 3.7700 | 0.0095 | 99.9305 |
| TFS-I | 98.5426 | 93.6727 | 99.3219 | 96.4146 | 99.32 | 0.0068 | 0.0165 | 6.3898 | 0.0146 | 99.9235 |
| **TFS-U** | **99.6121** | **98.4395** | **99.6131** | **99.0228** | **99.61** | **0.0039** | **0.0039** | **1.5606** | **0.0039** | **99.9797** |

The reasons for the superiority of the subset TFS-U are the combination of more comprehensive features. It contained a wider and more diverse set of features that reflect different aspects of the data, making it more compatible with the dynamic nature of IoMT. This was positively reflected in the model's superior ability to distinguish between normal and attack patterns.

To demonstrate how each model component affects performance, an ablation study was conducted to compare five model configurations:

Scenario 1: Processing raw data using DAE and DNN without feature selection (raw Features).

Scenario 2: Selecting the top 50% of features using IG-only with DAE and DNN (TFSIG).

Scenario 3: Retain the top 50% of features with DAE and DNN using RFE-only (TFSRFE).

Scenario 4: IG and RFE Integrating using the intersection of selected features with DAE and DNN (TFS-I).

Scenario 5: IG and RFE Integrating using the union of selected features with DAE and DNN (TFS-U).

As shown in Tables 7 and 8, the last scenario (TFS-U) showed a clear superiority in performance compared to the other scenarios. For example, accuracy increased to 99.93% using the WUSTL-EHMS-2020 dataset, compared to 90.196% using IG-only (TFSIG), and FAR decreased to 0.0007. This highlights the enhanced benefits of integrating these techniques into the model.

To assess the statistical reliability of the proposed model, ten independent experiments were conducted using the WUSTL-EHMS-2020 dataset. The proposed model achieved 99.823% average accuracy, a standard deviation of ±0.161, and a 95% confidence interval of ranging accuracy from 99.47% to 99.96%. As well as a one-sample t-test [43] against the baseline value of 99.5% returned a p-value of 0.00014, proving a statistically significant improvement is not because random variation.

Based on the study [44], which refers to the importance of inference latency and Floating Point Operations FLOPs, this work analyzed the approximate response time per sample (latency) and number of calculations (FLOPs) using the WUSTL-EHMS-2020 dataset. The model showed an inference latency of 112 ms for all features, decreasing significantly with feature selection: 70 ms (TFSIG), 68 ms (TFSRFE), 52 ms (TFS-I), and 59 ms (TFS-U). The total FLOPs for the model were estimated at 2,129 calculations.

According to the study [20], models achieving <100ms and <10,000 FLOPs are suitable for limited devices, and our model achieves 52–59ms and 2129 FLOPs, confirming its efficiency for deployment on Raspberry Pi and Jetson Nano.

Though the same models in other literature often have one performance measure, uniform performance runs attest to the robustness of the proposed model.

The Confusion matrices were created for each method utilizing DNN to assess the effectiveness of the proposed approaches, as depicted in Figs 8 and 9. The confusion matrix (A) displays the efficacy of the DNN using the original features. Matrices (B), (C), (D), and (E) indicate the effectiveness of DNN detection using TFSIG, TFSRFE, TFS-I, and TFS-U, respectively after dimension reduction by DAE based on both datasets.

Each matrix represents classification performance on various feature subsets: Original features, TFSIG, TFSRFE, TFS-I, and TFS-U. The classification algorithm should correctly identify more instances as true positive or true negative belonging to the class BENIGN or ATTACK, respectively. The matrix's diagonal elements provide the correctly classified numbers, and the elements off the diagonal reveal misclassifications. The accuracy obtained with minimum misclassification is shown with the maximum number of diagonal elements for the feature subset TFS-U.

Figs 10 and 11 show its training process on the WUSTL EHMS 2020 and CICIDS2017 datasets. As seen, accuracy moves up, while loss decreases, meaning this model learns well. Similarly, Figs 12 and 13 show the AUC for the two datasets, with a strong capability of the model to show good results in distinguishing benign from malignant traffic. The high AUC values obtained for both datasets proved that the proposed IDS model was robust and could be applied to extended IoMT environments.

The Proposed Model 99.9% and 99.6% accuracy at WUSTL-EHMS-2020 and CICIDS2017, respectively, indicate great promise of early cyberattack detection before they affect the connected medical devices. Remote patient monitoring systems can prevent service outages and data compromise by immediately detecting MitM or DDoS.

A light model and some carefully selected features reduce resource usage to render the model deployable on low-capacity devices like wearables. The model was trained and tested on heterogeneous, biometric signatures and network flows to capture the variability of IoMT device data and prove its versatility to varying operating environments.

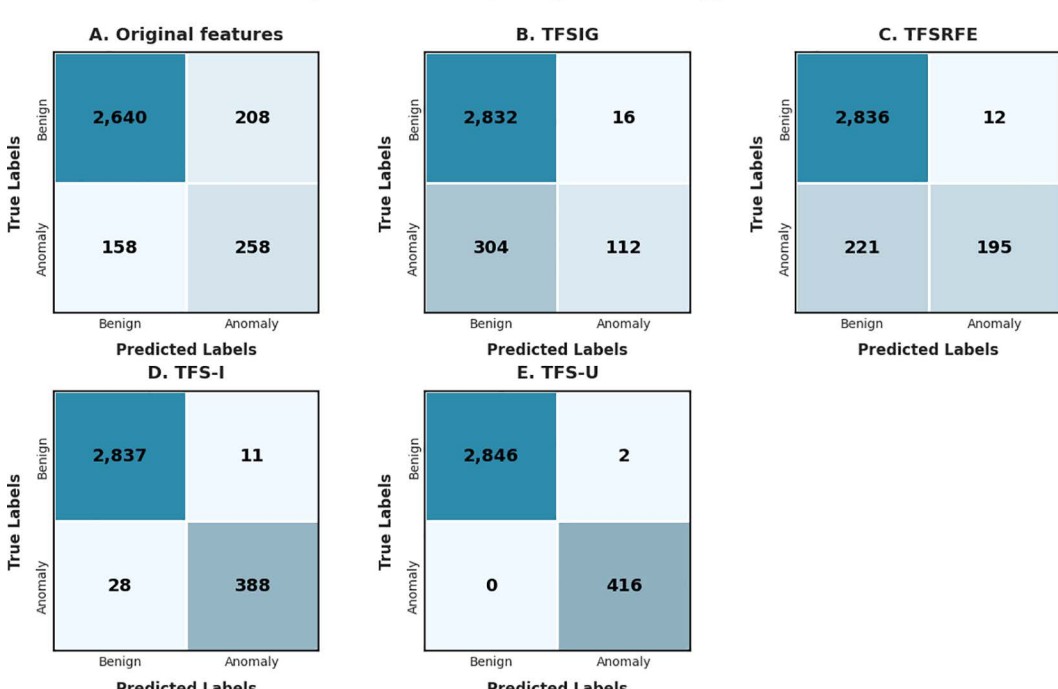

**Fig 8. Confusion Matrices based on the WUSTL-EHMS-2020 dataset.**

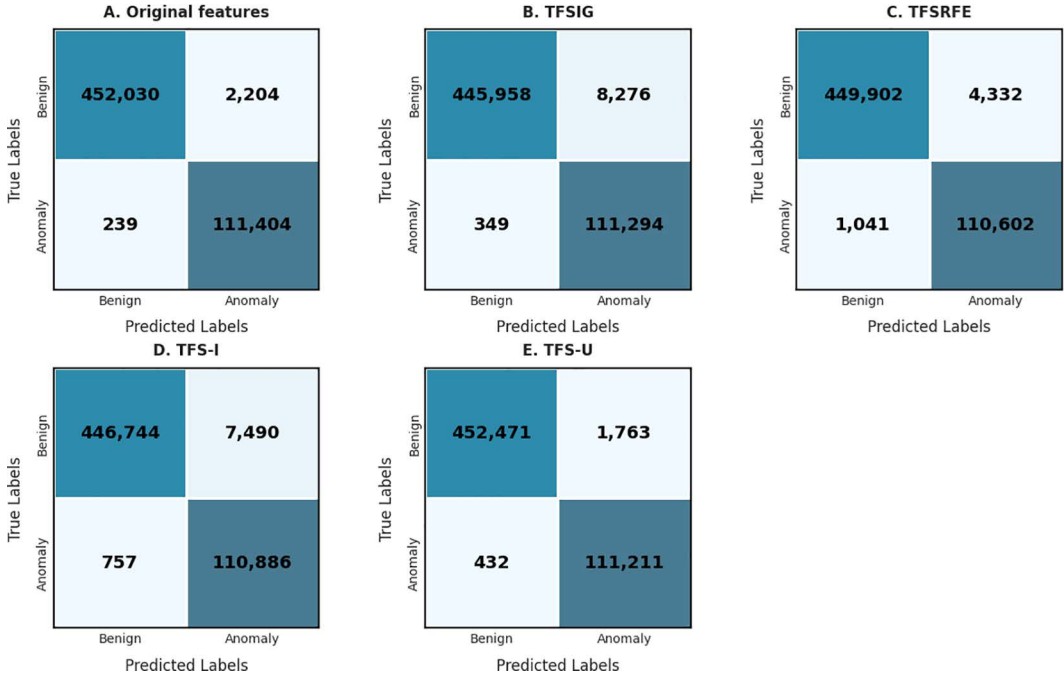

Fig 9. Confusion Matrices based on the CICIDS2017 dataset.

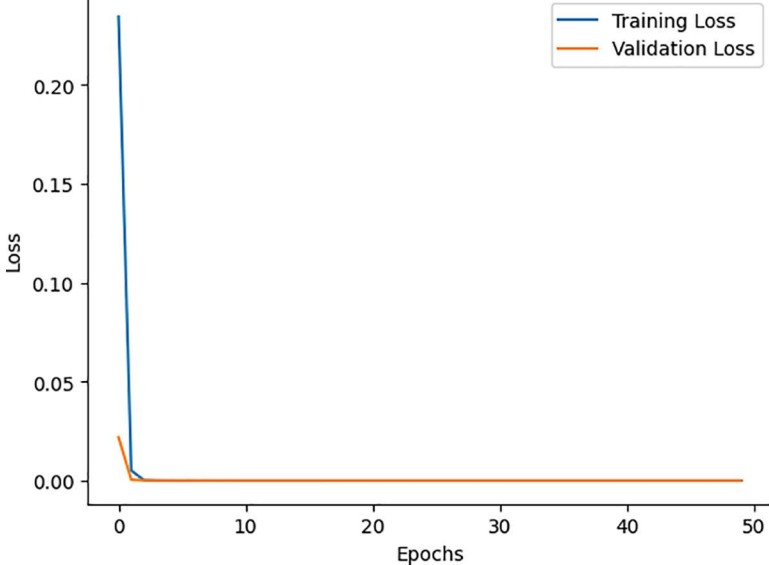

Fig 10. Represent the DNN training based on WUSTL EHMS 2020 dataset.

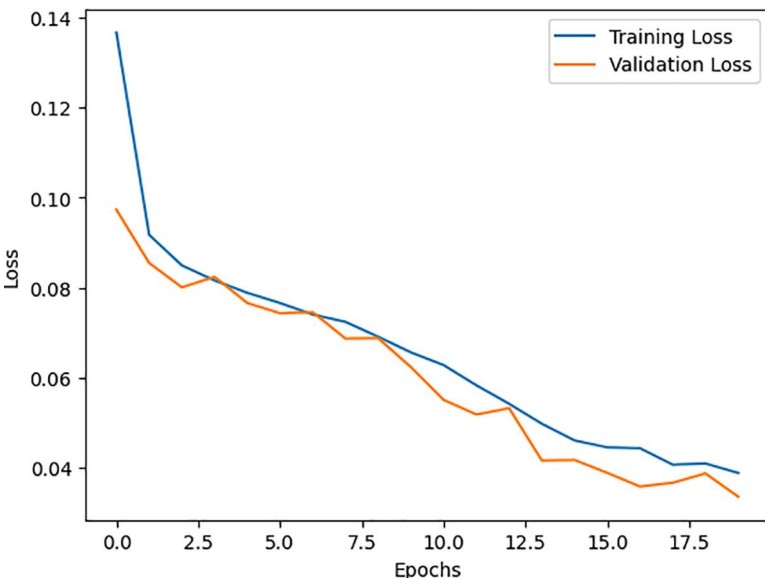

**Fig 11. Represent the DNN training based on the CICIDS2017 dataset.**

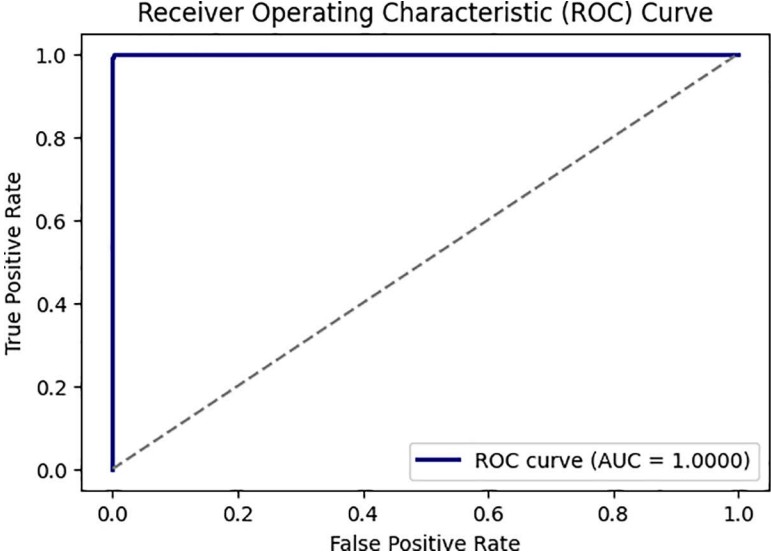

**Fig 12. Represent the DNN AUC based on WUSTL EHMS 2020 dataset.**

The proposed intrusion detection system model enhanced accuracy, memory consumption, and training time, but real-time latency and hardware-specific performance were not explicitly measured. The DNN architecture used in this study is relatively simple, using a small number of layers and nodes, and dimensionality reduction using DAE reduced the computational burden.

However, implementing low-power devices such as Raspberry Pi, microcontrollers presents processing, memory, and power consumption management issues. The approach has endpoint implementation potential; however, latency, CPU utilization, and power consumption must be tested. Future focus will be on these limitations.

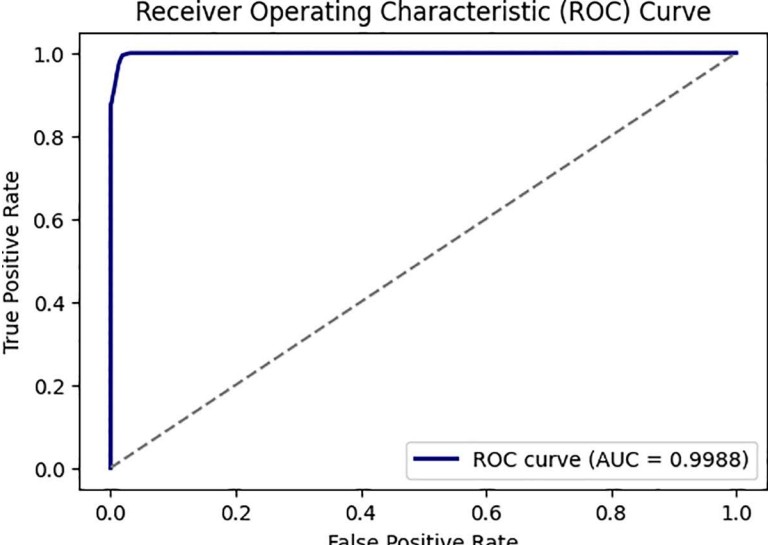

**Fig 13. Represent the DNN AUC based on the CICIDS2017 dataset.**

## Comparisons with recent models

As shown in Tables 9 and 10, the proposed approach's performance has been compared with state-of-the-art methods. The proposed model was evaluated against contemporary approaches in the IDS field using the WUSTL-EHMS-2020 and CICIDS2017 datasets. The proposed model demonstrated significant superiority, as seen by the findings presented in the mentioned Tables, as well as Figs 14 and 15.

Tables 9 and 10, and Figs 14, 15 compare the proposed model with the new state-of-the-art methods on the datasets of WUSTL-EHMS-2020 and CICIDS2017. The results obtained showcase that the proposed model performs better compared to other models in terms of achieved accuracy, precision, recall, and f1-score for both datasets. The model

**Table 9. Comparison with recent methods on the WUSTL-EHMS-2020 dataset.**

| Recent Studies | ACC % | PR% | REC% | F1% | AUC | MCC | T. Time/ Sec |
|---|---|---|---|---|---|---|---|
| [45] | 94.23 | 93.45 | – | 93.45 | 90.68 | – | 3.5 |
| [46] | 99 | 99 | 98 | 98 | – | – | 4.10 |
| [28] | 97.578 | 95.431 | 99.943 | 97.634 | – | – | – |
| [47] | 99.88 | 96 | – | 96.1 | – | – | – |
| [27] | 97.85 | 96.50 | 86.29 | – | 92.92 | 84.02 | – |
| **Proposed model** | **99.9387** | **99.5215** | **100** | **99.7602** | **100** | **99.59** | **3.2** |

**Table 10. Comparison with recent methods on the CICIDS2017 dataset.**

| Recent Studies | ACC % | PR% | REC% | F1% | AUC |
|---|---|---|---|---|---|
| [48] | 98 | 99 | 98 | 99 | 99 |
| [49] | 98.61 | 97.05 | 95.00 | 93.09 | – |
| [50] | 95.21 | 88.76 | 82,59 | 84.14 | – |
| [51] | 97.90 | 96.05 | – | 96.10 | – |
| [52] | 99.34 | 99.34 | 99.34 | 99.34 | 99.97 |
| [53] | 99.60 | 98.48 | 98.92 | 98.70 | 94.92 |
| **Proposed model** | **99.6121** | **98.4395** | **99.6131** | **99.0228** | **99.9797** |

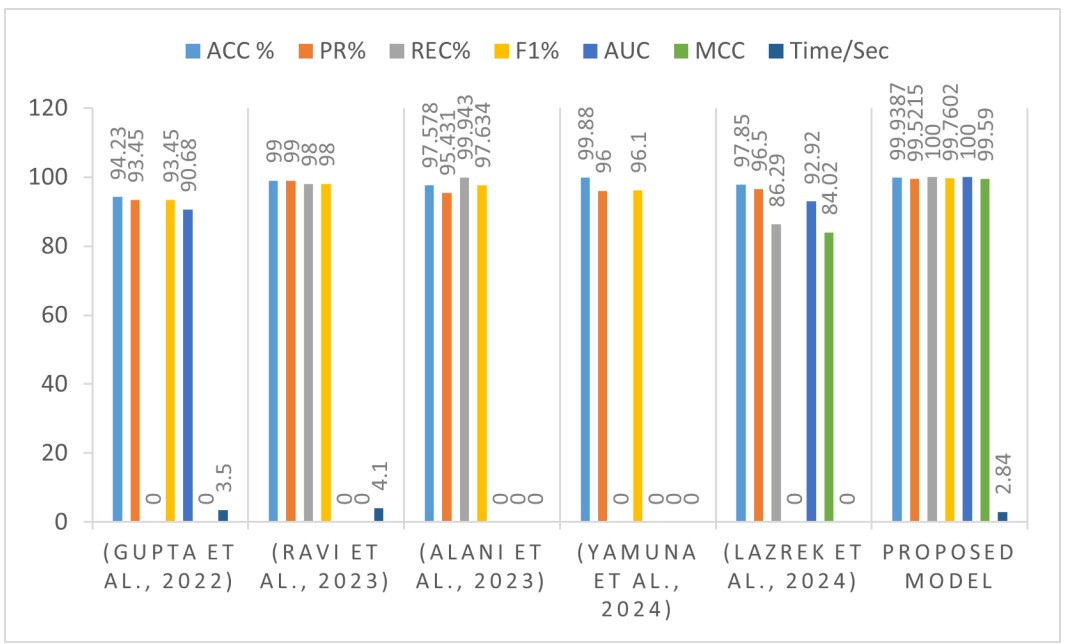

**Fig 14. Comparison of the proposed approach with recent studies using the WUSTL-EHMS-2020 dataset.**

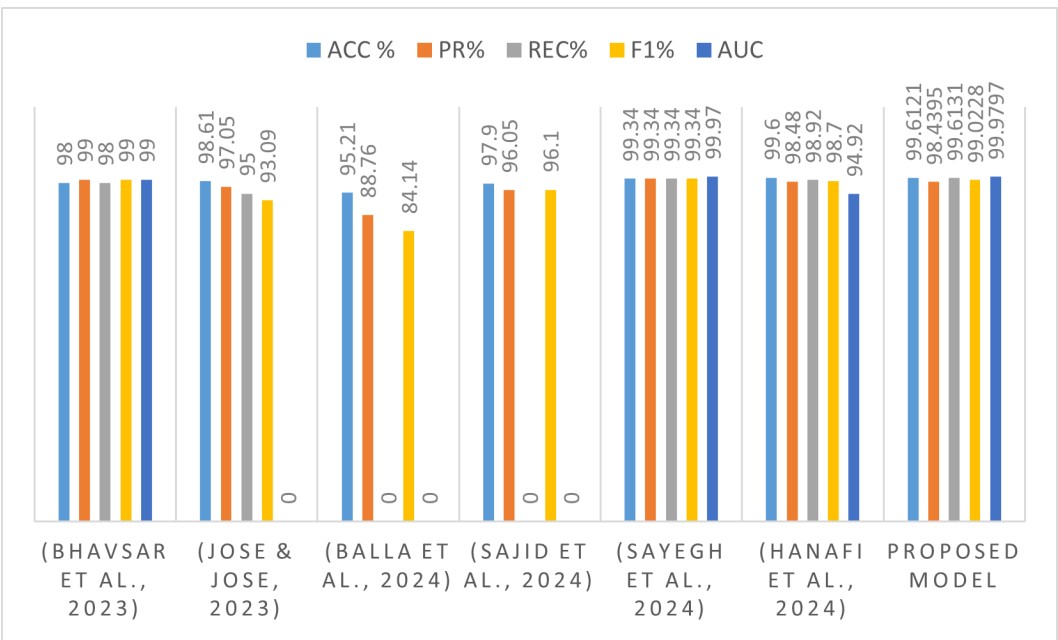

**Fig 15. Comparison of the approach's accuracy with recent studies using the CICIDS2017 dataset.**

achieved low misclassification and error rates with a significant reduction in memory consumption and training time, making it suitable for real-time use on limited IoMT devices. Its lightweight architecture and efficient feature selection make it highly suitable for embedded deployment in wearable or mobile medical devices. That explains why it performs better in discriminating between normal and malicious IoMT network traffic. The findings, like those of this study, prove that advanced feature selection methods do make an important contribution to deep-learning-model-based IDS working much better in IoMT settings.

This paper presents an analysis of memory utilization during feature selection to assess the efficacy of the suggested method in eliminating memory requirements. When utilizing the whole dataset (Original Features), the memory usage of WUSTL-EHMS-2020 and CICIDS2017 starts at 12.96 MB and 2.88 GB, respectively, as indicated in Table 11.

Nevertheless, the memory consumption significantly decreases when using TFSIG, TFSRFE, TFS-U, and TFS-I for feature selection, as seen in Table 11 variations in memory consumption during feature selection processes for two datasets, WUSTL-EHMS-2020 and CICIDS2017, are illustrated in Figs 16 and 17. These findings suggest that memory consumption significantly decreases during feature selection.

Unlike conventional IDS models that demand extensive computation, the proposed IDS for IoMT in this paper significantly reduces computational complexity while maintaining high accuracy, 99.93% and 99.61% for WUSTL-EHMS-2020 and CICIDS2017, respectively, as shown in Tables 7 and 8. Using the proposed IDS, memory consumption decreases from 12.96 MB to 4.11 MB for WUSTL-EHMS-2020 and from 2.88 GB to 1.16 GB for CICIDS2017, with training times

**Table 11. Memory Consumption Across Feature Sets in WUSTL-EHMS-2020 and CICIDS2017.**

| Memory Consumption | WUSTL-EHMS-2020 | CICIDS2017 |
|---|---|---|
| Original Features | 12.96 MB | 2.88 GB |
| TFSIG | 2.74 MB | 0.84 GB |
| TFSRFE | 2.74 MB | 0.84 GB |
| TFS-U (Union) | 4.11 MB | 1.16 GB |
| TFS-I (Intersection) | 1.37 MB | 0.53 GB |

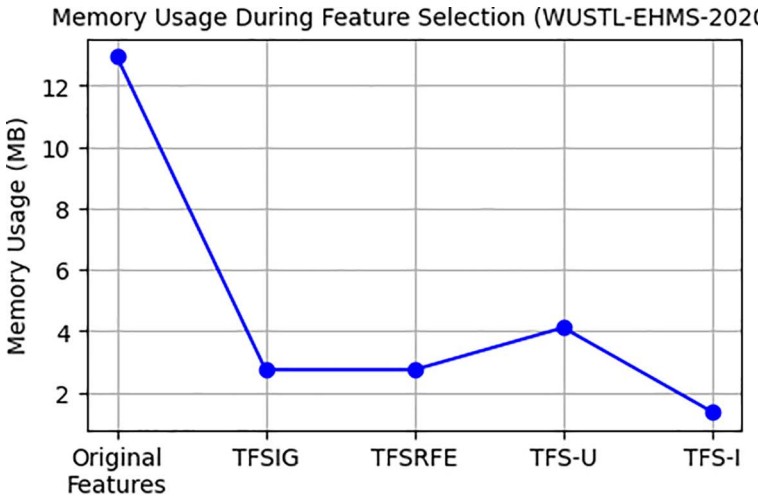

**Fig 16. Memory usage variation during feature selection from WUSTL-EHMS-2020.**

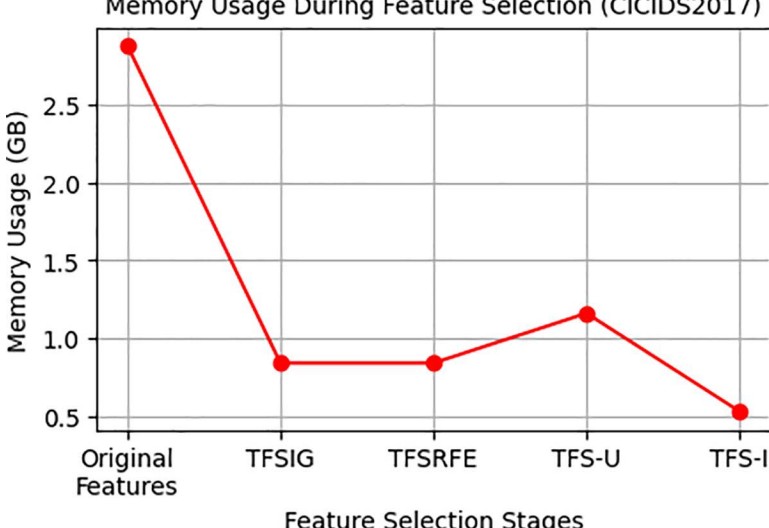

**Fig 17. Memory usage variation during feature selection from CICIDS2017.**

reduced from 12.84 to 3.2 seconds for WUSTL-EHMS-2020 as shown in Table 6. The model's low memory footprint and Low training time enable threat detection faster on resource-constrained IoMT devices [6]. This balance ensures reliable detection with minimal false negatives, while DAE enhances data privacy [40].

Compared to existing lightweight IDS like XMeDNN [28] and Ridge-CNN [27], proposed IDS exhibits superior accuracy while operating with reduced complexity. The proposed model has been carefully calibrated to maintain superior detection performance while achieving model consistency.

The proposed approach outperformed state-of-the-art approaches in accuracy, memory consumption, and training time. Confusion matrices and statistical analysis show that feature selection and dimensionality reduction enhanced model performance without losing accuracy. These findings demonstrate the model's efficiency, making it ideal for IoMT cyberse-curity systems.

## Conclusion

The paper proposes a new approach to enhancing the IDS of IoMT using advanced feature selection methods like IG and RFE, along with a DAE for dimensionality reduction. The model has been extensively evaluated using two independent datasets, WUSTL-EHMS-2020 and CICIDS2017, and demonstrated improved performance in terms of accuracy, preci-sion, and detection rate. Results show that the proposed IDS model reduces computational complexity and significantly enhances cyber threat detection in IoMT environments. This approach can address challenges related to involving high-dimensional data and threat detection, making it robust and resource-aware for both sensitive medical information protec-tion and healthcare service continuity. As shown in Tables 7, 8, and 11, the model decreased computational complexity by reducing training time and memory consumption. By taking advantage of reduced features, it achieved high accuracy while maintaining performance and computational efficiency.

## Future direction

The proposed model demonstrates good performance; however, it can be enhanced for resource-constrained IoMT environments. Future research will incorporate interpretability tools such as SHAP, LIME, or Grad-CAM to increase trans-parency and model trust. Experimental tests will be performed on devices such as Raspberry Pi or ESP32 to determine

real-time feasibility in clinical environments. It will include testing on real medical devices with compliance with IoMT security standards like HIPAA and GDPR. Advanced security measures like adaptive-based IDS, encryption, identity-based access control, and federated learning will be incorporated to ensure data privacy. The goal is to increase security, reduce memory consumption and increase scalability for deployment in resource-constrained environments. This plan will build trust in the system, determining its readiness for widespread use in healthcare settings.

## Author contributions

**Conceptualization:** Ahmed Muqdad Alnasrallah.

**Funding acquisition:** Ahmed Muqdad Alnasrallah.

**Investigation:** Ahmed Muqdad Alnasrallah.

**Methodology:** Ahmed Muqdad Alnasrallah.

**Project administration:** Ahmed Muqdad Alnasrallah, Maheyzah Md Siraj.

**Resources:** Ahmed Muqdad Alnasrallah, Hanan Ali Alrikabi.

**Software:** Ahmed Muqdad Alnasrallah.

**Supervision:** Maheyzah Md Siraj.

**Validation:** Ahmed Muqdad Alnasrallah, Maheyzah Md Siraj, Hanan Ali Alrikabi.

**Visualization:** Ahmed Muqdad Alnasrallah.

**Writing – original draft:** Ahmed Muqdad Alnasrallah.

**Writing – review & editing:** Ahmed Muqdad Alnasrallah, Hanan Ali Alrikabi.

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
