## [Decision Letter · Decision Letter 0]

19 Jan 2025

Dear Dr. Alnasrallah,

Thank you for submitting your manuscript to PLOS ONE. After careful consideration, we feel that it has merit but does not fully meet PLOS ONE’s publication criteria as it currently stands. Therefore, we invite you to submit a revised version of the manuscript that addresses the points raised during the review process.

We look forward to receiving your revised manuscript.

Kind regards,

mamoona humayun

Academic Editor

PLOS ONE

Journal Requirements:

Please confirm at this time whether or not your submission contains all raw data required to replicate the results of your study. Authors must share the “minimal data set” for their submission. PLOS defines the minimal data set to consist of the data required to replicate all study findings reported in the article, as well as related metadata and methods (https://journals.plos.org/plosone/s/data-availability#loc-minimal-data-set-definition ).

If your submission does not contain these data, please either upload them as Supporting Information files or deposit them to a stable, public repository and provide us with the relevant URLs, DOIs, or accession numbers. For a list of recommended repositories, please see https://journals.plos.org/plosone/s/recommended-repositories .

Additional Editor Comments:

Dear Authors,

Thank you for submitting your manuscript titled “Enhancing IDS for the IoMT based on Advanced Features Selection and Deep Learning Methods to Increase the Model Trustworthiness” to PLOS ONE.

After careful evaluation of the reviewers’ comments, I have decided that the manuscript requires Major Revision before further consideration. While one reviewer suggested minor revisions, the other recommended rejection. Considering the potential value of the work and the feedback provided, I believe the manuscript could be suitable for publication after addressing the reviewers’ concerns comprehensively.

Please revise your manuscript in accordance with the reviewers' comments and submit the updated version within 30 days.

Reviewers' comments:

Reviewer's Responses to Questions

**Comments to the Author**

1. Is the manuscript technically sound, and do the data support the conclusions?

Reviewer #1: Partly

Reviewer #2: Yes

2. Has the statistical analysis been performed appropriately and rigorously?

Reviewer #1: No

Reviewer #2: Yes

3. Have the authors made all data underlying the findings in their manuscript fully available?

Reviewer #1: Yes

Reviewer #2: Yes

4. Is the manuscript presented in an intelligible fashion and written in standard English?

Reviewer #1: No

Reviewer #2: Yes

Reviewer #1: The authors propose an IDS model for the IoMT that improves security and efficiency in the categorization of the detection process through the use of advanced feature selection approaches. Organization of the article is poor.

The figures are not embedded in the text which affects the readability of the text.

Even though the title says IoMT, all data sets and not IoMT datasets

Many abbreviations are used which are not expanded during their first use.

Methodology is not clear. Since the paper uses advanced feature selection process, comparison has to be done with reference to the features selected and the importance of these feature in the real life context. Result Analysis is also not satisfactory. Lot of grammatical errors are there. Thorough proofreading has to be done.

Reviewer #2: The abstract is well-written, and the topic is interesting. Provide a brief mention of numerical improvements (e.g., percentage increase in accuracy or F1 score).

Why were IG and RFE chosen over other feature selection methods? A brief rationale could strengthen the argument.

The use of Deep Autoencoder for dimensionality reduction is valid, but its specific advantage in this context needs clearer articulation.

Clarify how the proposed combination of IG, RFE, and Deep Autoencoder differs from or improves upon prior works.

Highlight why selecting the top 50% of features is optimal.

While the datasets are mentioned, their relevance to IoMT and the specific challenges they address are not fully explained. Why are these datasets suitable for evaluating IoMT IDS?

IDS and IoMT are defined again and again while they should be abbreviated at 1 time only when it is first used.

The description of methods like IG, RFE, and DAE lacks a deeper explanation of why these specific techniques were chosen over others. While their functionality is described, the rationale for their selection in the IoMT context is missing.

The section on Deep Neural Networks (DNNs) is generic and lacks details specific to the study, such as the architecture, hyperparameters, or training strategies used.

Discuss the scalability of these methods and their potential deployment in practical IoMT applications. Mention challenges such as data privacy, device compatibility, or computational constraints in real-time scenarios.

Highlight why the TFS-U subset outperformed other subsets in the CICIDS2017 and WUSTL-EHMS-2020 datasets.

Emphasize the novelty and advantages of combining DAE and DNN with selected features.

Discuss any challenges or limitations encountered in your study, such as computational costs, reliance on specific datasets, or potential overfitting.

Reflect on why certain feature subsets like TFSIG or TFSRFE underperformed relative to TFS-U.

While the model is stated to be suitable for real-time threat detection, there is no mention of potential latency or resource constraints.

Although the model is said to reduce computational complexity, no quantitative evidence is provided in the conclusion.

Specific directions for improvement, such as integrating more diverse datasets, optimizing the model for resource-constrained environments, or addressing new threat types, are missing

**Do you want your identity to be public for this peer review?** For information about this choice, including consent withdrawal, please see our Privacy Policy

Reviewer #1: No

Reviewer #2: No

---

## [Author Response · Author response to Decision Letter 1]

23 Feb 2025

Dear Professors.

We would like to thank the editorial board and reviewers for their insightful comments on the paper. Your feedback greatly improved the paper's clarity and research depth. Below are responses to each comment received, indicating the points of revision that are modified according to your values recommendations. We hope the amended work will meet scientific expectations.

Comments of the Esteemed Reviewer 1 and Our Responses:

1. Organization of the article is poor.

Response: We appreciate this feedback which is improving our research.

We have reorganized the article structure from abstract to references and focused on the Methodology section to make it more streamlined and clear.

2. The figures are not embedded in the text which affects the readability of the text.

Response: We thank the reviewer for his valuable feedback

Following the journal's guidelines, we attached the figures independently and not within the text. However, we have taken care to indicate the locations of appropriate figures within the text to enhance understanding of the text.

We welcome any additional comments to improve the paper.

3. Even though the title says IoMT, all datasets are not IoMT datasets.

Response: We appreciate your feedback on the depth of the result analysis.

CICIDS2017 is a comprehensive network dataset that encompasses attack patterns, such as DDoS and Man-in-the-Middle, commonly found in IoMT scenarios.

WUSTL-EHMS-2020 is specific to the Internet of Medical Things, encompassing biometric and network data from medical devices. This makes its combined use optimal for assessing general and healthcare-related threats.

To clarify this, we have explicitly explained its relevance in the dataset description section.

4. Many abbreviations are used which are not expanded during their first use.

Response: We have carefully reviewed and ensured that all abbreviations are expanded upon their first mention.

5. Methodology is not clear. Since the paper uses advanced feature selection process, comparison has to be done with reference to the features selected and the importance of these feature in the real life context.

Response: We appreciate your feedback regarding the clarity of the methodology.

In the revised manuscript, we have provided a detailed explanation of the methodology, especially the advanced feature selection process. Figure 1 has been included to visually illustrate the steps involved, helping to clarify the flow and rationale behind our approach. Where the suggested changes have been incorporated in the Proposed Model section after the title of Figure 1. And the following are our justifications:

Figure 1 after updated.

Clarification of Feature Selection Process:

The methodology follows a structured process that begins with selecting relevant features from the raw data, followed by applying Information Gain (IG) and Recursive Feature Elimination (RFE) as feature selection techniques.

The IG method was chosen for its ability to quantify the relevance of individual features by measuring their contribution to the target variable. This helps in identifying which features are most useful for classification tasks. On the other hand.

The RFE was employed to iteratively eliminate less important features, ensuring that only the most relevant ones remain for training the model. This combination of methods ensures that we not only select the most informative features but also reduce potential overfitting by eliminating redundant or irrelevant features.

IG and RFE were selected due to their ease of implementation and efficacy in managing the intricate datasets utilised in the study (e.g., CICIDS2017 and WUSTL-EHMS-2020), while maintaining a balance between correctness, complexity, and minimising computational expenses. These approaches maintain the interpretability of the original features and facilitate real-time applications in IoMT contexts.

The selection of the top 50% of features was based on empirical validation, optimizing both model performance and computing efficiency. This method preserved elevated accuracy while diminishing complexity, which is essential for IoMT applications. Cross-validation validated optimal performance with this ratio, reducing overfitting risks and ensuring robustness.

We have already added more clarification to the Proposed Model section and modified Figure1 according to your comments.

6. Result Analysis is also not satisfactory.

We sincerely appreciate the reviewer’s valuable feedback.

Response: We have provided comparisons with state-of-the-art studies, as shown in Tables 8 and 9, and included detailed performance evaluations using multiple metrics such as accuracy, Precision, recall, f1 score, detection rate, error rate, and memory consumption, as shown in Tables 6, 7, 10, and 11. We also added confusion matrices, Figures 8 and 9, and performance charts, Figures 14 and 15, in support of such depth in analysis. We acknowledge that we must strengthen the interpretation of our results, and for this.

We enriched the discussion better to bring out the practical benefits of the proposed model.

7. Lot of grammatical errors are there. Thorough proofreading has to be done.

Response: Comprehensive proofreading has been conducted to improve grammar and readability.

Comments of the Esteemed Reviewer 2 and Our Responses:

1. The abstract is well-written, and the topic is interesting. Provide a brief mention of numerical improvements (e.g., percentage increase in accuracy or F1 score).

Response: We appreciate your suggestion.

The proposed model achieves 99.61% accuracy with (CICIDS2017) and 99.93% accuracy with (WUSTL-EHMS-2020), surpassing state-of-the-art methods by 2.1% and 1.5%, respectively, while reducing training time as shown in Table 7 with WUSTL-EHMS-2020.

we added these details to the Abstract according to your suggestion.

2. Why were IG and RFE chosen over other feature selection methods? A brief rationale could strengthen the argument.

Response: We highly appreciate your deep feedback

IG and RFE have been selected because they are relatively simple to apply, and efficient in processing complex datasets that will be used here, such as CICIDS2017 and WUSTL-EHMS-2020, providing the best balance among accuracy, complexity, and reduction of computational costs. In addition, the approaches preserve the interpretability of the original features and enable their straightforward application within the IoMT contexts.

We have already added more clarifications to the Proposed Model section according to your comment.

3. The use of Deep Autoencoder for dimensionality reduction is valid, but its specific advantage in this context needs clearer articulation.

Response: We appreciate your deep feedback

DAE was chosen for its ability to learn nonlinear representations of high-dimensional data while preserving critical information, which reduces noise and improves model efficiency in an IoMT environment, where the data is feature-rich (e.g., biometric and network data), DAE helps reduce dimensionality without losing essential features, making the model lighter and faster to train and execute and increases the model’s focus on the patterns most relevant for classification. This effectiveness has been demonstrated experimentally by reduced memory usage (Tables 9 and 10, Figures 16 and 17) and enhanced performance in attack detection.

We highlighted that in the DAE section after Fig.4.

4. Clarify how the combination of IG, RFE, and Deep Autoencoder improves upon prior works.

Response: We acknowledge the importance of your feedback.

RFE, IG, and DAE were chosen due to their effectiveness in dealing with security challenges in IoMT environments. IDS suffers from high data dimensions and the need to achieve accurate threat detection with record response time. IG was adopted as a fast and effective filtering method to identify the most influential features without increasing the computational load, while RFE works to reduce overfitting by removing unnecessary features iteratively. As for DAE, it was adopted as an effective tool that showed a high ability to reduce dimensions while preserving the important information of the features, outperforming other methods, which was positively reflected in enhancing the accuracy of the model. Based on the tests we conducted while applying the proposed model on two datasets (CICIDS2017 and WUSTL-EHMS-2020), it outperformed in achieving 99.61% accuracy with CICIDS2017 and 99.93% with WUSTL-EHMS-2020, thus surpassing modern methods in terms of accuracy, detection rate, and reduced training time. Therefore, the selection of IG, RFE, and DAE can be considered as the optimal choice for computational accuracy, performance speed, and effective threat detection in IoMT systems.

5. Highlight why selecting the top 50% of features is optimal.

Response: We evaluate your important feedback

The selection of the top 50% of features was based on empirical validation, optimizing both model performance and computing efficiency. This method preserved elevated accuracy while diminishing complexity, which is essential for IoMT applications. Cross-validation validated optimal performance with this ratio, reducing overfitting risks and ensuring robustness.

6. While the datasets are mentioned, their relevance to IoMT and the specific challenges they address are not fully explained. Why are these datasets suitable for evaluating IoMT IDS?

CICIDS2017: Encompasses several network attacks (e.g., DDoS and Botnet), assessing the model's capacity to generalize to typical IoT settings.

WUSTL-EHMS-2020: focuses on healthcare-related assaults (e.g., data injection and identity theft), highlighting distinct IoMT concerns such as the sensitivity of medical data and the diversity of devices.

The combination of both datasets guarantees a thorough assessment of the model in practical situations.

7. IDS and IoMT are defined again and again while they should be abbreviated at 1 time only when it is first used.

Response: Thank you so much for your feedback

We have reviewed and ensured that each abbreviation is defined only once.

8. The section on Deep Neural Networks (DNNs) is generic and lacks details specific to the study, such as the architecture, hyperparameters, or training strategies used.

Response: We would like to thank you for suggesting that the explanation of the DNN section.

The DNN model in this study is designed to fulfill the requirements of IDS in IoMT, considering the problems of high dimensions and complexity of data. Three hidden layers of 64, 32, and 16 neurons using ReLU activation handle non-linear input. This number of neurons was chosen to reduce computational complexity and extract key information. The last layer consists of one neuron with the Sigmoid activation function for binary classification (BENIGN or ATTACK). The learning rate was set to 0.001 with the Adam optimizer, and batch size was 256, both carefully tuned in order to balance performance and complexity. Early Stopping and Reduce Learning Rate on Plateau enhanced model stability and reduced overlearning. This architecture provides perfect accuracy for two datasets.

We have added more details to the DNN section and explained how the model and its components are designed, the section has become more detailed and comprehensive.

9. Discuss the scalability of these methods and their potential deployment in practical IoMT applications. Mention challenges such as data privacy, device compatibility, or computational constraints in real-time scenarios.

Response: We appreciate the reviewer’s valuable feedback and insightful comments,

The proposed model is scalable and compatible with IoMT computational challenges, so many connected medical devices (e.g. wearable sensors) can be added using best feature selection techniques (IG, RFE) and dimensionality reduction techniques (DAE), which reduce computational limitations. Scalability and working with resource-constrained devices are possible due to the low memory utilization (Figures 16 and 17) and execution time (3.2 seconds).

Challenges that the model may face are: It may need to be fine-tuned when applied to another dataset because it relies on two standard datasets for training, compatibility with different devices, and privacy risks. Therefore, future work will focus on adaptive learning techniques and hybrid models to address and solve these problems.

We have already added more clarifications to the sections according to your comment.

10. Highlight why the TFS-U subset outperformed other subsets. Reflect on why subsets like TFSIG or TFSRFE underperformed relative to TFS-U.

Response: We are grateful for this valuable comment,

The reasons for the superiority of the TFS-U (Union of Top Feature Subsets) subset is the combination of more comprehensive features as it contains a wider and more diverse set of features that reflect different aspects of the data, which is positively reflected in the model's ability to distinguish between natural and attack patterns. Reducing information loss as a subset such as TFS-I (intersection) may exclude some important features that are not present in both intersecting subsets (TFSIG, TFSRFE) such as features related to uncommon attacks, while TFS-U maintains the diversity of features. TFSIG and TFSRFE suffer from limitations such as an apparent bias towards individual selection criteria especially in dynamic IoMT environments, which may lead to neglecting features with an indirect effect.

The experimental results and comparison of the model with recent studies showed the clear superiority of the model with the TFS-U subset, as appeared in tables 6,7,8,9.

We highlighted that in the Results and Measurements section after Table 7.

11. Emphasize the novelty and advantages of combining DAE and DNN with selected features.

Response: We are grateful for this valuable comment,

Based on practical experience and some previous studies, it has been proven that the combination of DAE and DNN with the selected features has a positive impact on the design of IDS in IoMT environments, as the proposed approach demonstrated accuracy in classification and efficiency in work, as it clearly shows superior results, reduced computational complexity, reduced memory consumption, reduced error rates, accelerated training time, stability and overcoming overfitting, as well as the possibility of expansion and protection of sensitive data because reducing dimensions via DAE reduces the risk of accessing all private data, which reduces the risk of hacking.

I added Emphasize accounting to your valuable comment in the Proposed Model section after Fig. 5.

12. While the model is stated to be suitable for real-time threat detection, there is no mention of potential latency or resource constraints.

Response: We thank you for your valuable comments that helped improve the quality of the paper.

The proposed model relied on advanced feature selection techniques (IG and RFE) to focus on the most useful features and leave the others. The model also benefited from DAE to reduce the dimensions of the selected features, which was reflected in significantly improving efficiency and accuracy, as the model achieved an accuracy of 99.61% with CICIDS2017 and 99.93% with WUSTL-EHMS-2020. Also, choosing the appropriate techniques for the model led to reducing the training time to 3.2 seconds. Memory consumption was also significantly reduced, which may make it compatible with devices with limited resources. The model also outperforms metrics such as AUC and MCC compared to recent methods, which indicates its robustness and effectiveness. These capabilities position the approach as an important step toward enhancing the security of IoMT networks. Future work will focus on calculating and improving latency to meet the requirements of critical applications and ensure compatibility with time-sensitive IoMT environments.

13. Although the model is said to reduce computational complexity, no quantitative evidence is provided in the conclusion.

Response: We appreciate the reviewer for this great comment.

We fully concur with your observation regarding the necessity of presenting quantifiable proof to substantia

---

## [Decision Letter · Decision Letter 1]

1 Apr 2025

Dear Dr.  Alnasrallah,

Thank you for submitting your manuscript to PLOS ONE.  After carefully considering the reviewers’ assessments, we are requesting a major revision of your manuscript to address the concerns outlined below.

**Clarity in Structural Improvements:** While you have stated that the article has been reorganized, there is no explicit mention of how readability, logical flow, and coherence have been enhanced. Please provide a clear summary of the structural improvements, detailing specific changes made to improve the article's presentation.**Dataset Justification:** The use of CICIDS2017 as a representative dataset for IoMT environments remains unconvincing, as it is a general IoT dataset. Please provide a more robust justification for its relevance to IoMT or consider using or supplementing with an IoMT-specific dataset.**Feature Selection and Practical Significance:** Your response to feature selection enhancements focuses on theoretical justification but lacks an explicit comparison with alternative methods. Additionally, a deeper discussion on the practical impact of the selected features in real IoMT deployments is necessary.

Please submit your revised manuscript by May 16 2025 11:59PM. If you will need more time than this to complete your revisions, please reply to this message or contact the journal office at plosone@plos.org . Please include the following items when submitting your revised manuscript:

We look forward to receiving your revised manuscript.

Kind regards,

mamoona humayun

Academic Editor

PLOS ONE

**Comments to the Author**

Reviewer #2: All comments have been addressed

Reviewer #3: (No Response)

2. Is the manuscript technically sound, and do the data support the conclusions?

Reviewer #2: (No Response)

Reviewer #3: Partly

3. Has the statistical analysis been performed appropriately and rigorously?

Reviewer #2: (No Response)

Reviewer #3: No

4. Have the authors made all data underlying the findings in their manuscript fully available?

Reviewer #2: (No Response)

Reviewer #3: Yes

5. Is the manuscript presented in an intelligible fashion and written in standard English?

Reviewer #2: (No Response)

Reviewer #3: Yes

Reviewer #2: (No Response)

Reviewer #3: 1. The authors state that they have reorganized the article.

Issue: There is no explicit mention of how the structure was improved. The response is vague and lacks specific details on how readability, logical flow, or coherence has been enhanced.

2. The authors justify CICIDS2017 by stating that it contains attacks relevant to IoMT environments.

Issue: CICIDS2017 is still a general IoT dataset, not an IoMT-specific one. Their justification does not fully address why a non-IoMT dataset is suitable.

3. They reviewed and expanded abbreviations at first mention.

Issue: This needs verification. If certain abbreviations remain undefined upon first usage, they need to correct this systematically.

4. They added details on IG and RFE selection, as well as the impact of feature selection.

Issue: The response focuses on theoretical justification but does not compare the selected features with alternative feature selection methods or explain the practical significance of these features in a real IoMT deployment. An explicit comparison with alternative selection methods and a deeper justification of why the chosen features matter in an IoMT-specific security context is needed to be included.

5. They expanded on the discussion, added confusion matrices, and included additional tables.

Issue: There is still insufficient discussion on practical implications, statistical significance of improvements, and how these findings translate into real-world benefits for IoMT networks. The authors should provide deeper insights into what these results mean in practical IoMT deployments—how do they handle adversarial attacks, real-time performance in constrained environments, and device heterogeneity?

6. They mentioned improvements in execution time and memory efficiency.

Issue: The response lacks a detailed latency analysis for real-time scenarios and does not quantify computational overhead. The authors should at least include numerical latency measurements, resource utilization data, and a discussion of the challenges in deploying the model on low-power IoMT devices.

7. They provided reduced training time and memory consumption statistics.

Issue: The provided numbers do not compare their approach with alternative methods or justify trade-offs. Reducing complexity while retaining model effectiveness in real-time applications is critical. Comparisons with other lightweight IDS models and a justification for why this specific trade-off is optimal are needed.

8. They added general directions for future research.

Issue: The proposed future work remains superficial. It lacks a concrete roadmap addressing:

How their model can be optimized for new IoMT attack types.

How they plan to validate performance in real IoMT environments.

Potential integration with real-world medical devices and compliance with IoMT security standards (e.g., HIPAA, GDPR).

**Do you want your identity to be public for this peer review?** For information about this choice, including consent withdrawal, please see our Privacy Policy

Reviewer #2: No

Reviewer #3: No

---

## [Author Response · Author response to Decision Letter 2]

29 Apr 2025

Dear Editor and Reviewers,

We sincerely thank you for your thorough and insightful feedback, which has significantly helped us improve the quality and clarity of our manuscript. We have carefully addressed each comment by incorporating additional details, restructuring sections, and providing deeper justifications to enhance the scientific rigor and practical relevance of our work. Below, we outline our responses to each comment, detailing the revisions made to the manuscript, including new paragraphs, tables, and analyses, to ensure alignment with your recommendations. We believe these changes strengthen the manuscript and better highlight its contributions to IoMT cybersecurity.

1. The authors state that they have reorganized the article.

Issue: There is no explicit mention of how the structure was improved. The response is vague and lacks specific details on how readability, logical flow, or coherence has been enhanced.

ANS:

We thank the reviewer for their valuable comments. We have reorganized the manuscript to improve the logical flow and coherence of the content by providing a clear roadmap for readers. To enhance clarity, the following paragraph has been added to the end of the introduction section to guide readers through the research structure:

"The remainder of this paper is structured as follows: Section 2 is a comprehensive review of related work. Section 3 describes the proposed IDS model framework with data preprocessing, feature selection, dimensionality reduction, and classification. Section 4 provides an overview of the experimental setup and datasets. Section 5 reports and discusses the evaluation results. Section 6 compares the proposed model with state-of-the-art approaches. Section 7 concludes the study. Finally, Section 8 provides future directions."

2. The authors justify CICIDS2017 by stating that it contains attacks relevant to IoMT environments.

Issue: CICIDS2017 is still a general IoT dataset, not an IoMT-specific one. Their justification does not fully address why a non-IoMT dataset is suitable.

ANS:

We thank the reviewers for their insightful comments.

We fully agree with the reviewer that the CICIDS2017 dataset is not exclusively rel for IoMT.

Therefore, WUSTL-EHMS-2020 was used as the primary reference because it represents a real IoMT environment.

CICIDS2017 was used as a supplementary dataset because it includes attacks relevant to IoMT environments.

Notably that several recent studies in the field of IoMT have also utilized the CICIDS2017 dataset as a reference for evaluating intrusion detection systems, given its variety of attacks and relevance to the behavior of IoMT.

For example, (Manimurugan et al.,2020) and (Balhareth & Ilyas, 2024) used CICIDS2017 to evaluate the performance of IDS in the IoMT context.

The following paragraph has been added to the end of the (Dataset Overview) section to explain the reason behind selecting each dataset and to reference studies that have used CICIDS2017 for the same reason:

" This study uses WUSTL-EHMS-2020, which includes biometric and network data from medical devices, for IoMT. CICIDS2017, a comprehensive dataset that encompasses attacks across various IoT scenarios, was utilized to broaden the evaluation. By integrating the two datasets, the model can generalize across contexts and test the system's ability to respond to complex and diverse challenges. Recent research like [36] and [37] used the CICIDS2017 dataset to evaluate IoMT security systems, supporting this methodology."

3. They reviewed and expanded abbreviations at first mention.

Issue: This needs verification. If certain abbreviations remain undefined upon first usage, they need to correct this systematically.

ANS:

We thank the reviewer for their valuable comments. All abbreviations are defined in Table 1.

Table 1: List of Abbreviations and Symbols.

Abbreviation Full Definition Abbreviation Full Form/Definition

IoT Internet of Things IoMT Internet of Medical Things

IDS Intrusion Detection System IG Information Gain

RFE Recursive Feature Elimination DAE Deep Autoencoder

DNN Deep Neural Network WUSTL-EHMS-2020 Washington University in St. Louis Electronic Health Monitoring System 2020 Dataset

ML Machine Learning CICIDS2017 Canadian Institute for Cybersecurity Intrusion Detection System 2017 Dataset

DL Deep Learning ACC Accuracy

PRE Precision DR Detection Rate

FNR False Negative Rate FPR/FAR False Positive Rate / False Alarm Rate

ICI Incorrect Classification Instances ER Error Rate

MCC Matthew’s Correlation Coefficient AUC Area Under the Curve

TP True Positive TN True Negative

FP False Positive FN False Negative

ReLU Rectified Linear Unit (activation function) TFSIG Top Fifty Percent-Feature Subset based on Information Gain

TFS-U Union of TFSIG and TFSRFE TFSRFE Top Fifty Percent-Feature Subset based on Recursive Feature Elimination

TFS-I Intersection of TFSIG and TFSRFE HIPAA Health Insurance Portability and Accountability Act (US data privacy regulation)

DoS Denial of Service GDPR General Data Protection Regulation (EU data privacy regulation)

DDoS Distributed Denial of Service MitM Man-in-the-Middle Attack

PSO Particle Swarm Optimization

4. They added details on IG and RFE selection, as well as the impact of feature selection.

Issue: The response focuses on theoretical justification but does not compare the selected features with alternative feature selection methods or explain the practical significance of these features in a real IoMT deployment. An explicit comparison with alternative selection methods and a deeper justification of why the chosen features matter in an IoMT-specific security context is needed to be included.

ANS:

(The following paragraph was added to the "Features Selection and Reduction Approach" section)

Thank you for this insightful comment. We agree that theoretical justifications alone are insufficient, so we will provide a more in-depth comparison between different methods and select the most relevant features for the medical environment, as shown below:

" Recent studies show that feature selection strategies improve IDS efficiency and accuracy in IoT contexts. Effective model complexity, accuracy, and training time reduction are essential for resource-constrained IoT systems that require fast and dependable responses. Similarly, the study [27], combining RFE-RIDGE with only 13 important features, improved classification performance to 97.85% accuracy and 0.03 false alarm rate. Reduced data dimensionality enhanced computational performance and complexity. The study [28] suggested IGRF-RFE, a hybrid technique for choosing key features. Selecting 23 of 42 UNSW-NB15 characteristics boosted classification accuracy to 84.24%. These approaches have increased computing efficiency and reduced complexity while preserving IDS performance. In [8], the authors presented LRGU-MIFS feature selection. This method calculates feature recurrence, decreasing feature duplication. Only 30 dataset features were picked using this method. This method improves performance and computational complexity, achieving 95.5% classification accuracy in IoMT.

Considering these results, the innovative approach to this work, which combines IG and RFE, using intersection and union methods, and DAE for dimensionality reduction, is effective and applicable in IoMT contexts. It balances high performance and computational efficiency while lowering feature size to the smallest set without compromising model accuracy. These features, including biometric and network flow data, are critical for detecting IoMT-specific threats such as spoofing or data injection"

5. They expanded on the discussion, added confusion matrices, and included additional tables.

Issue: There is still insufficient discussion on practical implications, statistical significance of improvements, and how these findings translate into real-world benefits for IoMT networks. The authors should provide deeper insights into what these results mean in practical IoMT deployments—how do they handle adversarial attacks, real-time performance in constrained environments, and device heterogeneity?

ANS:

We sincerely thank the reviewer for this valuable and constructive feedback. Based on this valuable feedback, we have made the following modifications to the manuscript to clarify the practical aspects further:

i. Statistical significance of improvements:

(The following paragraph was added to the "Experimental Results and Analysis" section after Table 7.)

To assess the statistical reliability of the proposed model, ten independent experiments were conducted using the WUSTL-EHMS-2020 dataset. The proposed model achieved 99.823% average accuracy, a standard deviation of ±0.161, and a 95% confidence interval of ranging accuracy from 99.47% to 99.96%. As well as a one-sample t-test [40] against the baseline value of 99.5% returned a p-value of 0.00014, proving a statistically significant improvement is not because random variation.

Though the same models in other literature often have one performance measure, uniform performance across runs attests to the robustness of the proposed model.

ii. Real-World Practical Implications, Device Heterogeneity and Adversarial Attacks: Based on this valuable revision, we expand the Discussion section to show how our results assist IoMT networks:

(The following paragraph was added to the end of the "Experimental Results and Analysis" section after Fig.13)

the Proposed Model 99.9% and 99.6% accuracy at WUSTL-EHMS-2020 and CICIDS2017, respectively, indicate great promise of early cyberattack detection before they affect the connected medical devices. Remote patient monitoring systems can prevent service outages and data compromise by immediately detecting MitM or DDoS.

A light model and some carefully selected features reduce resource usage to render the model deployable on low-capacity devices like wearables. The model was trained and tested on heterogeneous, biometric signatures and network flows to capture the variability of IoMT device data and prove its versatility to varying operating environments.

iii. Real-Time Performance:

We have reformulated part of the conclusion to illustrate the model's suitability for IoMT environments with performance and resource constraints as follows:

"The model achieved low misclassification and error rates with a significant reduction in memory consumption and training time, making it suitable for real-time use on limited IoMT devices. Its lightweight architecture and efficient feature selection make it highly suitable for embedded deployment in wearable or mobile medical devices."

6. They mentioned improvements in execution time and memory efficiency.

Issue: The response lacks a detailed latency analysis for real-time scenarios and does not quantify computational overhead. The authors should at least include numerical latency measurements, resource utilization data, and a discussion of the challenges in deploying the model on low-power IoMT devices.

ANS:

We thank the reviewer for his constructive feedback. We fully agree on the importance of including an analysis of latency and computational overhead, especially when targeting real-time applications and resource-constrained devices in an IoMT environment.

In this research, we focused on improving performance by reducing model complexity using feature selection techniques (IG and RFE) and dimensionality reduction using DAE, which resulted in significant memory consumption reductions and reduced training time, as shown in Tables 6, 7, 10, and 11.

However, we did not conduct direct measurements of response time or resource consumption at the hardware level, and we acknowledge that this is a limitation of the current work.

Our main goal was to reduce model complexity using feature selection techniques (IG and RFE) and dimensionality reduction using DAE.

These improvements indicate a suitable possibility for deployment on lightweight and resource-constrained devices.

i. In response to your valuable comments, we have enhanced the Discussion section by adding an explanation of the challenges of deployment on low-power IoMT devices, as follows:

"The proposed intrusion detection system model enhanced accuracy, memory consumption, and training time, but real-time latency and hardware-specific performance were not explicitly measured. The DNN architecture used in this study is relatively simple, using a small number of layers and nodes, and dimensionality reduction using DAE reduced the computational burden.

However, implementation on low-power devices such as Raspberry Pi, microcontrollers presents processing, memory, and power consumption management issues. The approach has endpoint implementation potential, however, latency, CPU utilization, and power consumption must be tested. Future focus will be on these limitations."

ii. A paragraph has also been added in the Future Work section confirming our intention to conduct a comprehensive evaluation of response time and computing load on real platforms to ensure the model is ready for practical deployment.

"The proposed model has shown superior performance, but to ensure sustainable performance in resource-constrained and critical IoMT environments, it can be integrated with more comprehensive datasets to cover a wider range of uncommon or advanced attacks scenarios, addressed by adaptive, hybrid learning, and reinforcement learning strategies, and evaluated for latency, computational, and feasibility. Experimental work with Raspberry Pi or ESP32 to better understand the approach's practical uses in time-sensitive medical applications."

7. They provided reduced training time and memory consumption statistics.

Issue: The provided numbers do not compare their approach with alternative methods or justify trade-offs. Reducing complexity while retaining model effectiveness in real-time applications is critical. Comparisons with other lightweight IDS models and a justification for why this specific trade-off is optimal are needed.

ANS:

We thank the reviewer for this insightful comment. In response to this comment, we have revised the manuscript to include explicit comparisons with other lightweight intrusion detection models, both in terms of training time and memory consumption, as shown in Tables 8–10 and Figures 16–17.

In addition, we have added a new paragraph to the Discussion section to explain the rationale for the balance between reducing complexity and maintaining performance in the proposed model design.

Our reliance on a combination of feature selection techniques (IG and RFE) and dimensionality reduction using DAE allows for a highly compact representation without losing accuracy—and even achieving higher performance. This balance makes the model well-suited for IoMT environments.

Below are the modifications we made to the paper in response to your valuable comments.

Memory Consumption WUSTL-EHMS-2020 CICIDS2017

Original Features 12.96 MB 2.88 GB

TFSIG 2.74 MB 0.84 GB

TFSRFE 2.74 MB 0.84 GB

TFS-U (Union) 4.11 MB 1.16 GB

TFS-I (Intersection) 1.37 MB 0.53 GB

Table 10. Memory Consumption Across Feature Sets in WUSTL-EHMS-2020 and CICIDS2017.

"Unlike conventional IDS models that demand extensive computation, the proposed IDS for IoMT in this paper significantly reduces computational complexity while maintaining high accuracy, 99.93% and 99.61% for WUSTL-EHMS-2020, and CICIDS2017 respectively, as shown in Tables 6 and 7. Using the proposed IDS, memory consumption decreases from 12.96 MB to 4.11 MB for WUSTL-EHMS-2020 and from 2.88 GB to 1.16 GB for CICIDS2017, with training times reduced from 12.84 to 3.2 seconds for WUSTL-EHMS-2020 as shown in Table 6. The model’s low memory footprint and Low training time enable threat detection faster on resource-constrained IoMT devices [6]. This balance ensures reliable detection with minimal false negatives, while DAE enhances data privacy [37].

Compared to existing lightweight IDS like XMeDNN [25] and Ridge-CNN [24], proposed IDS exhibits superior accuracy while operating with reduced complexity. The proposed model has been carefully calibrated to maintain superior detection performance while achieving model consistency. "

8. They added general directions for future r

---

## [Decision Letter · Decision Letter 2]

14 May 2025

Dear Dr. Alnasrallah,

Thank you for submitting your manuscript to PLOS ONE. After careful consideration, we feel that it has merit but does not fully meet PLOS ONE’s publication criteria as it currently stands. Therefore, we invite you to submit a revised version of the manuscript that addresses the points raised during the review process.

We look forward to receiving your revised manuscript.

Kind regards,

mamoona humayun

Academic Editor

PLOS ONE

Journal Requirements:

Reviewers' comments:

Reviewer's Responses to Questions

**Comments to the Author**

Reviewer #2: All comments have been addressed

Reviewer #3: (No Response)

2. Is the manuscript technically sound, and do the data support the conclusions?

Reviewer #2: Yes

Reviewer #3: Partly

3. Has the statistical analysis been performed appropriately and rigorously?

Reviewer #2: Yes

Reviewer #3: Yes

4. Have the authors made all data underlying the findings in their manuscript fully available?

Reviewer #2: Yes

Reviewer #3: Yes

5. Is the manuscript presented in an intelligible fashion and written in standard English?

Reviewer #2: Yes

Reviewer #3: No

Reviewer #2: (No Response)

Reviewer #3: This manuscript proposes an intrusion detection system (IDS) framework for the Internet of Medical Things (IoMT), integrating Information Gain (IG) and Recursive Feature Elimination (RFE) for feature selection, Deep Autoencoder (DAE) for dimensionality reduction, and Deep Neural Network (DNN) for classification. While the authors have made significant improvements in this revision and addressed most reviewer suggestions, several areas still require clarification or further elaboration:

1. The abstract still includes high-level claims (e.g., “enhances detection efficiency”) but does not clearly describe the pipeline components (IG, RFE, DAE, DNN) or mention statistical significance tests.

Suggestion: Revise the abstract to explicitly outline the full model pipeline and include key statistical validation elements such as “99.82% ± 0.16 accuracy on WUSTL-EHMS-2020 with p < 0.001.”

2. The response outlines prior works and justifies IG + RFE, but the architectural innovation and integration rationale remain vague.

Suggestion: Include an ablation study comparing: (1) IG-only, (2) RFE-only, (3) IG+RFE, and (4) IG+RFE+DAE to demonstrate the performance gain from each module. Clarify why this combination is particularly beneficial in IoMT.

3. The manuscript does not incorporate or mention any interpretability tool (e.g., SHAP, LIME) or provide qualitative analysis.

Suggestion: Add a discussion on interpretability limitations and propose Grad-CAM or SHAP integration as future work for clinical usability. Interpretability is essential for trust in medical systems.

4. Authors acknowledge not performing hardware-specific latency analysis and only include memory and training time.

Suggestion: Report approximate inference latency per sample and FLOPs. Mention specific low-power devices (e.g., Raspberry Pi, Jetson Nano) and expected deployment feasibility. If empirical results are not available, cite similar works for estimation.

5. While the manuscript includes several foundational works on IoMT security and feature selection, it omits recent and relevant studies on lightweight intrusion detection, feature optimization strategies, and AI-based real-time threat analysis in IoT/IoMT environments. These omissions limit the depth and currency of the literature review.

Suggestion:

Incorporate and discuss the following peer-reviewed works to enhance the contextual framework and align the proposed IDS model with the current state-of-the-art in AI-driven cybersecurity:

Sana et al. (2024). Securing the IoT cyber environment: Enhancing intrusion anomaly detection with vision transformers. IEEE Access. https://doi.org/10.1109/ACCESS.2024.3404778

→ Demonstrates a modern transformer-based IDS architecture with strong performance on IoT data, offering a meaningful comparison point for your deep learning approach.

Dai et al. (2024). An intrusion detection model to detect zero-day attacks in unseen data using machine learning. PLoS ONE, 19(9), e0308469. https://doi.org/10.1371/journal.pone.0308469

→ Provides insight into generalization and robustness for detecting unknown threats, which is particularly relevant given your use of CICIDS2017.

Yee et al. (2024). A Systematic Literature Review on AI-Based Methods and Challenges in Detecting Zero-Day Attacks. IEEE Access, 12, 144150–144163. https://doi.org/10.1109/ACCESS.2024.3455410

→ Offers a comprehensive survey of AI approaches to zero-day detection and can support your justification for using hybrid and dimensionality-reducing models like IG+RFE+DAE.

He et al. (2024). SeizureLSTM: An optimal attention-based trans-LSTM network. Biomedical Signal Processing and Control, 96, 106603. https://doi.org/10.1016/j.bspc.2024.106603

→ Highlights architectural design choices involving attention and feature integration in time-sensitive medical contexts, offering transferable insights to IoMT IDS development.

Incorporating these references would enrich the literature review, strengthen the justification for the proposed hybrid model, and demonstrate alignment with cutting-edge research in AI-driven IoMT security.

6. The comparison focuses mostly on prior DNN-based models; non-DNN or transformer-based IDS are not addressed.

Suggestion: Expand the Related Work section to discuss transformer-based models (e.g., Vision Transformer IDS), SVM-based lightweight IDS, or federated IDS approaches. This contextualizes your model better within the broader research space.

7. Figures have been included, but clarity and resolution are still suboptimal. ROC curves and ablation tables are hard to read.

Suggestion: Re-render all figures at high resolution, ensure axis labels and legends are visible, and include a consolidated summary table comparing all feature subsets (IG, RFE, union, intersection) in one place.

8. There are redundant claims (e.g., high accuracy, low memory) across abstract, results, and conclusion.

Suggestion: Refactor the conclusion to emphasize insight over repetition. Avoid qualitative phrases like “perfect accuracy” without statistical evidence. Use “statistically significant improvement” instead.

**Do you want your identity to be public for this peer review?** For information about this choice, including consent withdrawal, please see our Privacy Policy

Reviewer #2: No

Reviewer #3: No

---

## [Author Response · Author response to Decision Letter 3]

27 May 2025

Dear Editor and Reviewers,

We sincerely thank you for your thorough and insightful feedback, which has sig-nificantly helped us improve the quality and clarity of our manuscript. We have carefully addressed each comment by incorporating additional details, restructur-ing sections, and providing deeper justifications to enhance the scientific rigor and practical relevance of our work. Below, we outline our responses to each comment, detailing the revisions made to the manuscript, including new para-graphs, tables, and analyses, to ensure alignment with your recommendations. We believe these changes strengthen the manuscript and better highlight its contribu-tions to IoMT cybersecurity.

1. The abstract still includes high-level claims (e.g., “enhances detection ef-ficiency”) but does not clearly describe the pipeline components (IG, RFE, DAE, DNN) or mention statistical significance tests.

ANS:

Thank you for your constructive feedback. We have revised the abstract to explicitly describe the model pipeline and the statistical validation components. Specifically, we have included pipeline components and key statistical validation metrics, as be-low:

" The proposed model demonstrates superior performance in terms of accuracy, pre-cision, recall, and F1 score. It achieves an accuracy of 99.93% on the WUSTL-EHMS-2020 dataset while reducing training time and attains 99.61% accuracy on the CICIDS2017 dataset. The model achieved an average accuracy of 99.82% ± 0.16% and a p-value of 0.0001 on the WUSTL-EHMS-2020 dataset, which refers to stable statistical improvement. This study shows that the proposed strategy decreases com-putational complexity and enhances IDS accuracy and efficiency in IoMT networks. "

This revision now ensures clarity and reflects the critical components of the method-ology as you suggested.

2. The response outlines prior works and justifies IG + RFE, but the archi-tectural innovation and integration rationale remain vague.

ANS:

Thank you for your valuable comments. We understand the need to clarify the rationale for innovation and architectural integration behind the proposed model. In response to your valuable suggestion, the discontinuity study has been included in the research after Table 8, as follows:

" As well as demonstrating how each model component affects performance, an ablation study was conducted to compare five model configurations:

Scenario 1: Processing raw data using DAE and DNN without feature selec-tion (raw Features).

Scenario 2: Selecting the top 50% of features using IG-only with DAE and DNN (TFSIG).

Scenario 3: Retain the top 50% of features with DAE and DNN using RFE-only (TFSRFE).

Scenario 4: IG and RFE Integrating using the intersection of selected features with DAE and DNN (TFS-I).

Scenario 5: IG and RFE Integrating using the union of selected features with DAE and DNN (TFS-U).

As shown in Tables 7 and 8, the last scenario (TFS-U) showed a clear superi-ority in performance compared to the other scenarios. For example, accuracy increased to 99.93% using the WUSTL-EHMS-2020 dataset, compared to 90.196% using IG-only (TFSIG), and FAR decreased to 0.0007. This high-lights the enhanced benefits of integrating these techniques into the model."

3. The manuscript does not incorporate or mention any interpretability tool (e.g., SHAP, LIME) or provide qualitative analysis.

ANS:

We sincerely thank the reviewer for highlighting the importance of interpret-ability in medical systems and for suggesting tools like SHAP, LIME, or Grad-CAM to enhance the trustworthiness model. We completely agree that interpretability is a critical aspect of deploying machine learning models in healthcare settings, where understanding the rationale behind predictions can significantly improve user confidence and adoption.

To address this concern, we have updated the "Future Direction" section of the manuscript to explicitly discuss the limitations of interpretability in the current work and propose the integration of advanced interpretability tools as part of future research.

4. Authors acknowledge not performing hardware-specific latency analysis and only include memory and training time.

ANS:

We thank the reviewer for pointing out the importance of analyzing latency and FLOPs as part of evaluating the computational performance of the model. Based on this observation, we conducted additional experiments to evaluate latency and FLOPs using the WUSTL-EHMS-2020 datasets. As shown in the following text, which was added to the research:

"Based on the study [41], which refers to the importance of inference latency and Floating Point Operations FLOPs, this work analyzed the approximate re-sponse time per sample (latency) and number of calculations (FLOPs) using the WUSTL-EHMS-2020 dataset. The model showed an inference latency of 112 ms for all features, decreasing significantly with feature selection: 70 ms (TFSIG), 68 ms (TFSRFE), 52 ms (TFS-I), and 59 ms (TFS-U). The total FLOPs for the model were estimated at 2,129 calculations.

According to the study [42], models achieving <100ms and <10,000 FLOPs are suitable for limited devices, and our model achieves 52–59ms and 2129 FLOPs, confirming its efficiency for deployment on Raspberry Pi and Jetson Nano."

5. While the manuscript includes several foundational works on IoMT secu-rity and feature selection, it omits recent and relevant studies on light-weight intrusion detection, feature optimization strategies, and AI-based real-time threat analysis in IoT/IoMT environments. These omissions lim-it the depth and currency of the literature review.

ANS:

We thank the reviewer for his important observation regarding the compre-hensiveness and modernity of the references in the Related Work section. We fully agree with the importance of strengthening the literature review by in-corporating recent studies relevant to IoMT. Based on the recommendation, the suggested studies have been included in the section as shown below:

"Recent literature, which focuses on innovative approaches to enhanced accuracy and operational efficiency, provides a deeper understanding of IDS in IoMT environments. For example, a study [20] presented an IDS model based on Vision Transformers, demonstrating how modern deep learning techniques can be used to efficiently process complex data. The research [21] focused on developing models capable of detecting zero-day attacks using machine learning techniques, reflecting the importance of generalization when dealing with unknown or unexpected data. Furthermore, the study [22] conducted a comprehensive review and detailed analysis of the challenges associated with using artificial intelligence to detect zero-day attacks, recommending the use of hybrid models and dimensionality reduction techniques to improve performance.

Finally, the study [23] demonstrated the potential for designing networks with attention-based architectures to achieve better integration of time-sensitive features, providing valuable insights applicable to highly sensitive IoMT en-vironments. "

6. The comparison focuses mostly on prior DNN-based models; non-DNN or transformer-based IDS are not addressed.

ANS:

We thank the reviewer for his valuable comments on the importance of ex-panding the literature discussion to IDS models based on other techniques. This is discussed in the Related Works section, as shown below:

" The research [21] focused on developing models capable of detecting zero-day attacks using machine learning techniques, reflecting the importance of generalization when dealing with unknown or unexpected data. "

7. Figures have been included, but clarity and resolution are still subopti-mal. ROC curves and ablation tables are hard to read.

ANS:

We thank the reviewer for their valuable comments on the clarity of the illus-trations and the quality of the presentation. We recognize that the quality of the images plays a critical role in accurately communicating research results. Based on the recommendation:

The graphs (ROC curves and confusion matrices) have been redesigned with higher resolution, with improved clarity of the axes and legends, and axis la-bels.

The colors and display styles have been standardized across all graphs for consistency and ease of understanding.

In addition, Tables 7 and 8 have been updated to summarize the model per-formance results based on different feature sets (IG, RFE, intersection, and union).

8. There are redundant claims (e.g., high accuracy, low memory) across the abstract, results, and conclusion.

ANS:

We thank the reviewer for his careful observation regarding the repetition of some phrases throughout the Abstract, Results, and Conclusion sections. The article has been rewritten some parts, based on your insightful comments.

We are deeply grateful for the constructive feedback provided by the reviewers, which has guided us in refining our manuscript to meet the highest standards of aca-demic rigor and practical applicability. We hope that the revised manuscript meets your expectations. We remain open to any further suggestions to ensure the quality of the final publication.

Sincerely,

---

## [Editor Report · Decision Letter 3]

11 June 2025

Enhancing IDS for the IoMT based on Advanced Features Selection and Deep Learning Methods to increase the model trustworthiness

PONE-D-24-51421R3

Dear authors,

We’re pleased to inform you that your manuscript has been judged scientifically suitable for publication and will be formally accepted for publication once it meets all outstanding technical requirements.

Kind regards,

mamoona humayun

Academic Editor

PLOS ONE

**Comments from PLOS Editorial Office** : We note that one or more reviewer has recommended that you cite specific previously published works in the current and previous rounds of revision. As always, we recommend that you please review and evaluate the requested works to determine whether they are relevant and should be cited. It is not a requirement to cite these works and you may remove any added citations before the manuscript proceeds to publication. We appreciate your attention to this request.
---

## [Editor Report · Acceptance letter]

PONE-D-24-51421R3

PLOS ONE

Dear Dr. Alnasrallah,

I'm pleased to inform you that your manuscript has been deemed suitable for publication in PLOS ONE. Congratulations! Your manuscript is now being handed over to our production team.

Kind regards,

on behalf of

Dr. mamoona humayun

Academic Editor

PLOS ONE